# A Unified Theory of Sinusoidal Activation Families for Implicit Neural Representations

## Abstract

Implicit Neural Representations (INRs) model continuous signals with compact neural networks and have become a standard tool in vision, graphics, and signal processing. A central challenge is accurately capturing fine detail without heavy hand-crafted encodings or brittle training heuristics. Across the literature, periodic activations have emerged as a compelling remedy: from SIREN, which uses a single sinusoid with a fixed global frequency, to more recent architectures employing multiple sinusoids and, in some cases, trainable frequencies and phases. We study this *family* of sinusoidal activations and develop a principled theoretical and practical framework for trainable sinusoidal activations in INRs. Concretely, we instantiate this framework with **S**inusoidal **T**rainable **A**ctivation **F**unctions (**STAF**), a Fourier-series activation whose amplitudes, frequencies, and phases are learned. Our analysis (i) establishes a Kronecker-equivalence construction that expresses trainable sinusoidal activations with standard sine networks and quantifies expressive growth, (ii) characterizes how the Neural Tangent Kernel (NTK) spectrum changes under trainable sinusoidal parameterization, and (iii) provides an initialization that yields unit-variance post-activations without asymptotic central limit theorem (CLT) arguments. Empirically, on images, audio, shapes, inverse problems (super-resolution, denoising) and NeRF, STAF is competitive and often superior in reconstruction fidelity, with consistently faster early-phase optimization. While periodic activations can alleviate *practical manifestations* of spectral bias, our results indicate they do not eliminate it; instead, trainable sinusoids reshape the optimization landscape to improve the *capacity–convergence* trade-off.

## 1 Introduction

Implicit Neural Representations (INRs) approximate continuous signals by mapping coordinates to signal values via multilayer perceptrons (MLPs). INRs underpin modern view synthesis, 3D geometry, and compact signal representations (Mildenhall et al., 2020; Rahaman et al., 2019; Sitzmann et al., 2020; Tancik et al., 2020; Liu et al., 2024a; Kazerouni et al., 2024; Saragadam et al., 2023). A recurring observation is that standard ReLU networks exhibit a training-time preference toward low frequencies (Rahaman et al., 2019), which complicates faithful recovery of high-detail content unless aided by positional encodings or architectural changes.

A parallel thread replaces or augments activations themselves. SIREN (Sitzmann et al., 2020) showed that a global sinusoid dramatically improves fidelity, and follow-up work explored alternative periodic bases (e.g., Gabor-like (Saragadam et al., 2023)) and multi-sine constructions; some of these make the sinusoidal parameters trainable. Collectively, these results suggest that *sinusoidal activations* are a broad and useful family for INRs.

In this paper, we organize and extend this line of work by providing theory and practice for *trainable* sinusoidal activations. We instantiate the framework with **S**inusoidal **T**rainable **A**ctivation **F**unction (**STAF**), which represents each activation by a short Fourier series with learnable amplitudes, frequencies, and phases. Our goals are twofold: (i) **Theoretical grounding.** We formalize how trainable sinusoids increase effective frequency support and how this affects the Neural Tangent Kernel (NTK) spectrum and optimization dynamics. (ii) **Practical recipe.** We provide a layer-wise shared activation parameterization and a simple initialization producing unit-variance post-activations. **Contributions:**

- **Unified view of sinusoidal activations.** We position SIREN and subsequent multi-sinusoid variants within a common class and study *trainable* sinusoidal activations for INRs, clarifying when and why they help.

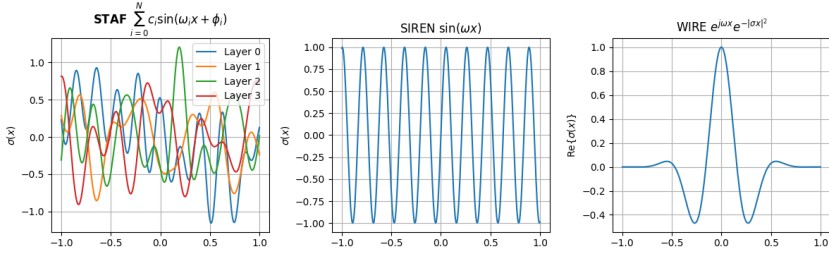

Figure 1: Activation functions used in INRs plotted over the range [-1, 1]. STAF utilizes a parameterized Fourier series activation, offering flexible frequency-domain adaptation. SIREN employs a sinusoidal function, providing a periodic activation landscape. WIRE employs a complex Gabor wavelet activation, balancing spatial and frequency localization.

- **Kronecker-equivalence & expressive growth.** We prove that networks with trainable sinusoidal activations admit an equivalent construction using plain sine activations with structured (Kronecker) weights. This yields quantifiable growth of the set of potential frequencies by a factor of $\tau^K$ (Theorems 4.2, 4.3).

- **Capacity–convergence analysis via NTK.** We analyze how trainable sinusoids reshape the NTK eigen-spectrum and relate larger leading eigenvalues to faster learning of specific components, providing criteria that connect activation parameters to convergence behavior.

- **Initialization without CLT assumptions.** We give an initialization that directly yields unit-variance post-activations for sinusoidal series (Theorem 3.1), removing dependence on distributional approximations.

- **Empirical study across tasks.** On images, audio, shapes, inverse problems, and NeRF, STAF is competitive and often better in PSNR/SSIM and convergence speed, with favorable parameter efficiency under layer-wise sharing. These advancements stem from faster convergence and enhanced accuracy, establishing STAF as a superior alternative to SOTA models, including INCODE (Kazerouni et al., 2024), FINER (Liu et al., 2024a), WIRE (Saragadam et al., 2023), SIREN (Sitzmann et al., 2020), Gaussian (Ramasinghe & Lucey, 2022), FFN (Tancik et al., 2020).

## 2 RELATED WORKS

INRs have been applied to a wide range of signals (including images, audio, signed distance fields (SDFs), and compressed representations) by fitting coordinate MLPs to continuous targets. Early work with sine activations demonstrated that periodic nonlinearities can substantially boost fidelity in coordinate-based modeling (Sitzmann et al., 2020). Broader architectural strategies further improved capacity and efficiency, including dual-MLP modulation (Mehta et al., 2021), dividing inputs into grids or ensembles (Aftab et al., 2022; Kadarvish et al., 2021), and adaptive allocation of model resources (Martel et al., 2021; Saragadam et al., 2022).

For 3D scenes, NeRF couples INRs with differentiable volume rendering (Mildenhall et al., 2020), inspiring numerous advances that trade off fidelity, speed, and memory. These include higher-quality rendering and antialiasing (Barron et al., 2023; Wu et al., 2023), factorized or accelerated pipelines (Chen et al., 2024; Reiser et al., 2021), Jacobian-aware or regularized training (Xu et al., 2023), fast super-resolution and compact encodings (Lin et al., 2024; Kazerouni et al., 2024), and extensions to more general or dynamic settings (Uy et al., 2024; Li et al., 2025).

Activation design has shaped neural network trainability and expressivity. Saturating sigmoids suffer from vanishing gradients, motivating unbounded or piecewise-linear alternatives such as ReLU and its variants (Nair & Hinton, 2010; Maas et al., 2013; Elfwing et al., 2018; Hendrycks & Gimpel, 2016). Trainable or adaptive activations (e.g., Swish, TanhSoft, SinLU) further tailor nonlinearity to data (Ramachandran et al., 2017; Biswas et al., 2021; Paul et al., 2022). In the INR setting, training dynamics interact with frequency content: ReLU-based networks tend to fit lower frequencies first (Rahaman et al., 2019). This observation motivated periodic activations, which embed high-frequency structure directly into the activation space. Although early attempts at periodic networks highlighted practical training difficulties (Lapedes & Farber, 1987; Parascandolo et al., 2016), subsequent INR work demonstrated stable recipes and strong reconstructions with sinusoidal activations and modulated architectures (Sitzmann et al., 2020; Mehta et al., 2021). More recently,

variable-periodic activations explicitly *tune* spectral bias in INRs (Liu et al., 2024a), and training protocols for sinusoidal networks have been refined for robustness and stability (Novello et al., 2025).

Beyond changing the activation itself, explicit frequency mappings inject sinusoidal structure at the input. Classical Fourier Neural Networks proposed trigonometric expansions inside the network (Gallant & White, 1988), while modern Fourier feature mappings provide a simple, widely adopted mechanism for enriching frequency support in INRs (Tancik et al., 2020). These approaches are complementary to periodic activations and are often combined in practice. Recent work explores alternative functional parameterizations for coordinate models, including Kolmogorov–Arnold Networks (KAN) and polynomial KAN variants (Liu et al., 2024b; SS et al., 2024). Such methods target expressive yet structured representations and offer a contrasting lens on building frequency-rich function classes for INRs.

## 3    A UNIFIED VIEW OF SINUSOIDAL ACTIVATIONS

We formulate *STAF* as a family of sinusoidal activation functions that *encompasses* prior INR activations, from SIREN's single fixed-frequency sine to multi-sine and trainable-frequency variants (Liu et al., 2024a; Novello et al., 2025). This framing enables us to study common mechanisms (expressivity, initialization, and optimization dynamics) without enumerating each prior design choice. For clarity and to avoid repetition, we, therefore, use *SIREN as the canonical base model* in comparisons and discussions, noting that it arises as a special case within our formulation.

### 3.1    INR PROBLEM FORMULATION

INRs employ MLPs as a method for representing continuous data. At the core of INR is the function $f_{\boldsymbol{\theta}} : \mathbb{R}^{F_0} \to \mathbb{R}^{F_L}$, where $F_0$ and $F_L$ represent the dimensions of the input and output spaces, respectively, and $\boldsymbol{\theta}$ denotes the parameters of the MLP. The objective is to approximate a target function $g(\boldsymbol{x})$ such that $g(\boldsymbol{x}) \approx f_{\boldsymbol{\theta}}(\boldsymbol{x})$. For example, in image processing, $g(\boldsymbol{x})$ could be a function mapping pixel coordinates to their respective values.

As mentioned in (Yüce et al., 2022), the majority of INR architectures can be decomposed into a mapping function $\gamma : \mathbb{R}^D \to \mathbb{R}^T$ followed by an MLP, with weights $\boldsymbol{W^{(l)}} \in \mathbb{R}^{F_l \times F_{l-1}}$ and activation function $\rho^{(l)} : \mathbb{R} \to \mathbb{R}$, applied element-wise at each layer $l = 1, \ldots, L - 1$. In other words, if we represent $\boldsymbol{z^{(l)}}$ as the post-activation output of each layer, most INR architectures compute

$$
\begin{aligned}
\boldsymbol{z^{(0)}} &= \gamma(\boldsymbol{r}), \\
\boldsymbol{z^{(l)}} &= \rho^{(l)}(\boldsymbol{W^{(l)}} \boldsymbol{z^{(l-1)}} + \boldsymbol{B^{(l)}}), \quad l = 1, ..., L - 1, \\
f_{\boldsymbol{\theta}}(\boldsymbol{r}) &= \boldsymbol{W^{(L)}} \boldsymbol{z^{(L-1)}} + \boldsymbol{B^{(L)}}.
\end{aligned}
\tag{1}
$$

Additionally, corresponding to the $i$'th neuron of the $l$'th layer, we employ the symbols $a_i^{(l)}$ and $z_i^{(l)}$ for the pre-activation and post-activation functions, respectively. The choice of the activation function $\rho$ is pivotal in INR, as it influences the network's ability to represent signals. Traditional functions, such as ReLU, may not effectively capture high-frequency components. The novel parametric periodic activation function, i.e., STAF, enhances the network's capability to accurately model and reconstruct complex, high-frequency signals.

### 3.2    STAF: SINUSOIDAL TRAINABLE ACTIVATION FUNCTION

We adopt a Fourier–series activation drawn from the broader family of sinusoidal activations used in INRs (see Figure 1):

$$
\rho^*(x) = \sum_{i=1}^{\tau} C_i \sin(\Omega_i x + \Phi_i),
\tag{2}
$$

where $C_i$, $\Omega_i$, and $\Phi_i$ represent the *amplitude*, *frequency*, and *phase* parameters, respectively. These parameters are learned dynamically during training, enabling the network to adapt its activation function to the specific characteristics of the signal being modeled. The use of a Fourier series is motivated by its ability to represent signals efficiently, capturing essential components with a small number of coefficients. This adaptability allows STAF to provide a compact and flexible representation for complex patterns in various tasks.

### 3.3    STAF TRAINING PROCESS

During training, STAF optimizes not only the traditional MLP parameters (weights and biases), but also the coefficients of the activation function. This dual optimization approach ensures that the

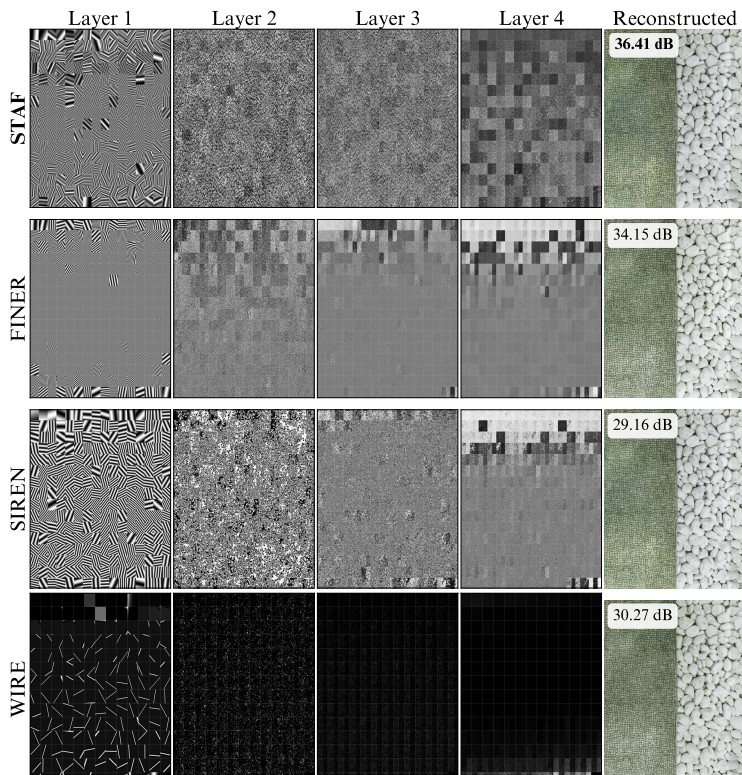

Figure 2: Activation maps of STAF, FINER, SIREN, and WIRE learned during image reconstruction.

network learns both an optimal set of transformations (through weights and biases) and an ideal way of activating neurons (through the parametric activation function) for each specific task. The training employs a reconstruction loss function designed to minimize the difference between the target function $g(\boldsymbol{x})$ and the network's approximation $f_{\boldsymbol{\theta}}(\boldsymbol{x})$, while also encouraging efficient representation inspired by Fourier series.

### 3.4 IMPLEMENTATION STRATEGIES

The implementation of STAF's parametric activation functions can be approached in three ways:

❶ **Individual Neuron Activation:** This method assigns a unique activation function to each neuron. It offers high expressiveness but leads to a significant increase in the number of trainable parameters, making it impractical for large networks due to potential overfitting and computational inefficiencies.

❷ **Uniform Network-wide Activation:** Here, a single shared activation function is used across the entire network. This approach simplifies the model by reducing the number of additional parameters but limits the network's expressiveness and adaptability. It may struggle to capture diverse patterns and details in complex signals.

❸ **Layer-wise Shared Activation:** This balanced strategy employs a distinct shared activation function for each layer, which is used for all experiments in this paper. For example, in a 3-layer MLP with $\tau = 25$ terms, only 225 additional parameters are required. This approach optimally balances expressiveness and efficiency, allowing each layer to develop specialized activation dynamics for the features it processes. It aligns with the hierarchical nature of MLPs, where different layers capture different signal abstractions, providing an efficient learning mechanism tailored to each layer's role.

### 3.5 INITIALIZATION

In this section, we present an initialization strategy tailored for networks utilizing STAF as the activation function. While STAF shares similarities with SIREN (Sitzmann et al., 2020), which employs sin as its activation function, our initialization scheme is specifically designed to leverage the unique parameterization of STAF. To provide context, we first revisit the key aspects of SIREN's initialization scheme as discussed in (Sitzmann et al., 2020), and then highlight how our approach builds upon and extends these principles to enhance network performance and stability.

In SIREN, the input $X$ of a single neuron follows a uniform distribution $U(-1, 1)$, and the activation function employed is $\rho(u) = \sin(u)$. Consequently, the output of the neuron is given by $Y = \sin(aX + b)$, where $a, b \in \mathbb{R}$. The authors of (Sitzmann et al., 2020) claim that regardless of the choice of $b$, if $a > \frac{\pi}{2}$, the output $Y$ follows an arcsine distribution, denoted as $Arcsine(-1, 1)$. However, it becomes apparent that this claim is not correct upon further examination. If the claim were true, the moments of $Y$ would be independent of $b$. However, this only occurs when $a = n\pi$. We have demonstrated this in the Appendix D.1.

In the subsequent parts of (Sitzmann et al., 2020), the authors assumed that the outputs of the first layer follow a distribution of *arcsine* and fed those outputs into the second layer. By relying on the central limit theorem (CLT), they demonstrated that the output of the second layer, for each neuron, conforms to a normal distribution. Additionally, in Lemma 1.6 (Sitzmann et al., 2020), they established that if $X \sim \mathcal{N}(0, 1)$ and $Y = \sin(\frac{\pi}{2}X)$, then $Y \sim Arcsine(-1, 1)$. However, it should be noted that to prove this result, they relied on several approximations. Through induction, they asserted that the inputs of subsequent layers follow an arcsine distribution, while the outputs of these layers exhibit a normal distribution.

In contrast to the approach taken by (Sitzmann et al., 2020), the method presented in this study does not depend on the specific distributions of the input vector $r$ and weight matrices $W^{(l)}$. As a result, there is no need to map the inputs to the interval $[-1, 1]$. Additionally, this method does not rely on making any approximations or the central limit theorem, which assumes large numbers. Overall, it offers a more rigorous mathematical framework. To pursue this goal, notice the following theorem.

**Theorem 3.1.** *Consider a neural network as defined in equation 1 with a sinusoidal trainable activation function (STAF) as in equation 2. For convenience, define $\Gamma_i = \Omega_i \, w \cdot x$. Suppose that for each $i$ we have: $\Phi_i \sim U(-\pi, \pi)$, and the random variables $C_i$ follow the probability density function*

$$f_{C_i}(c_i) = \frac{\tau |c_i|}{2} e^{-\frac{\tau c_i^2}{2}}. \tag{3}$$

*In addition, assume the following independence conditions:*

- *the variables $C_i$ are mutually independent;*

- *for each $i$, $C_i$ is independent of $(\Gamma_i, \Phi_i)$; and*

- *the collections $\{(C_i, \Gamma_i, \Phi_i)\}_{i=1}^{\tau}$ are mutually independent.[1]*

*Then, every post-activation will follow a $\mathcal{N}(0, 1)$ distribution (refer to the proof in Appendix D.2.)*

This initial setting, where every post-activation follows a standard normal distribution, is beneficial because it prevents the post-activation values from vanishing or exploding. This ensures that the signals passed from layer to layer remain within a manageable range, particularly in the first epoch, which establishes the foundation for subsequent learning (Yüce et al., 2022). If the learning process is well-posed and there is sufficient data, the training process is likely to converge to a stable and accurate solution. Therefore, while it is important to monitor potential issues in later epochs, the concern about vanishing or exploding values is significantly greater during the initial stages. Proper initialization helps mitigate these risks early, facilitating smoother and more effective training overall.

## 4 THEORY: EXPRESSIVITY, CAPACITY, AND CONVERGENCE

This section develops the theory behind trainable sinusoidal activations for INRs and states the main results used in the paper. (i) We begin with a *Kronecker-equivalence* construction (Theorem 4.2) showing how networks with trainable sinusoidal activations can be represented by sine networks with structured (Kronecker) weights; we use this to quantify the growth of potential frequencies via a Delannoy-number bound (Theorem 4.3). (ii) We then connect *capacity* to *convergence* by discussing how these activations reshape the NTK spectrum and what this implies for learning dynamics.

For readability, we present statements, intuition, and their consequences in the main text, while deferring full technical material to the Appendix: *(A)* the complete proof of Theorem 4.2 appears in

---

[1]Note that this condition does not necessarily imply internal independence within each triplet. In other words, the independence of the triplets does not imply that $C_i$ is independent of $(\Gamma_i, \Phi_i)$.

Appendix D.3; *(B)* the injectivity/perturbation argument used to preserve the size of the potential-frequency set is formalized in Lemma 4.4 with proof in Appendix D.4; *(C)* the closed-form dual activation and derivative for STAF, and the resulting NTK recursion, are derived in Appendix A (Theorem A.2), which also includes empirical NTK computation details and extended eigenvalue/eigenfunction visualizations (Fig. 4); to support that derivation, the Gaussian integral identities are proved in Appendix A.1.1; *(D)* the unit-variance initialization result referenced by our analysis has its proof in Appendix D.2. We use these ingredients to justify the capacity growth, explain the observed eigen-spectra, and clarify why periodic activations *mitigate* (rather than eliminate) practical manifestations of spectral bias.

Let us first examine the expressive power of our architecture, drawing upon the notable Theorem 1 from (Yüce et al., 2022). This theorem is as follows:

**Theorem 4.1.** *(Theorem 1 of (Yüce et al., 2022)) Let $f_{\boldsymbol{\theta}} : \mathbb{R}^D \to \mathbb{R}$ be an INR of the form of Equation equation 1 with $\rho^{(l)}(x) = \sum_{j=0}^{J} \alpha_j x^j$ for $l > 1$. Furthermore, let $\boldsymbol{\Psi} = [\boldsymbol{\Psi}_1, ..., \boldsymbol{\Psi}_T]^{tr} \in \mathbb{R}^{T \times D}$ and $\boldsymbol{\zeta} \in \mathbb{R}^T$ denote the matrix of frequencies and vector of phases, respectively, used to map the input coordinate $r \in \mathbb{R}^D$ to $\gamma(r) = \sin(\boldsymbol{\Psi}r + \boldsymbol{\zeta})$. This architecture can only represent functions of the form*

$$f_{\boldsymbol{\theta}}(r) = \sum_{\boldsymbol{w'} \in \mathcal{H}(\Psi)} c_{w'} \sin(\langle \boldsymbol{w'}, \boldsymbol{r} \rangle + \zeta_{\boldsymbol{w'}}),$$

*where*

$$\mathcal{H}(\boldsymbol{\Psi}) \subseteq \tilde{\mathcal{H}}(\boldsymbol{\Psi}) = \left\{ \sum_{t=1}^{T} s_t \boldsymbol{\Psi}_t \,\middle|\, s_t \in \mathbb{Z} \wedge \sum_{t=1}^{T} |s_t| \leq J^{L-1} \right\}.$$

Please note the following remarks regarding this theorem:

**Remark 5.1.1.** We refer to $\tilde{\mathcal{H}}$ as the set of potential frequencies.

**Remark 5.1.2.** The expression $\sum_{t=1}^{T} s_t \boldsymbol{\Psi}_t$ is equal to $\boldsymbol{\Psi}^{tr}[s_1, ..., s_T]^{tr}$. This representation is more convenient for our subsequent discussion, as we will be exploring the kernel of $\boldsymbol{\Psi}$ in the sequel.

**Remark 5.1.3.** In the context of SIREN, where $\rho^{(l)} = \sin$, the post-activation function of the first layer, $z^{(0)} = \sin(\omega_0(\boldsymbol{W}^{(0)}r + \boldsymbol{b}^{(0)}))$, can be interpreted as $\gamma(\boldsymbol{r}) = \sin(\boldsymbol{\Psi}r + \boldsymbol{\zeta})$.

We next investigate the expressive power of the proposed activation. To facilitate comparison with SIREN, we express our network using $\sin$ as the activation function.

Let us consider a neural network with a parametric activation function defined in equation 2. To represent our network using SIREN, we demonstrate that every post-activation function of our network from the second layer onwards ($z^{l+1}$) can be expressed using linear transformations and sine functions. Notably, the final post-activation function ($z^{(L-1)}$) can be constructed using SIREN, albeit requiring more neurons than STAF. In other words, our network can be described using a SIREN and some Kronecker products denoted by $\otimes$. This analysis resembles that provided in (Jagtap et al., 2022), with a slight difference in the settings of the paper. In (Jagtap et al., 2022), it was shown that an adaptive activation function of the form $\rho^*(x) = \sum_{i=1}^{\tau} C_i \rho_i(\Omega_i x)$ can be represented using a feed-forward neural network, where each layer has neurons with activation functions $\rho_i$. To align STAF with this theorem, we must have $\rho_i = \sin(\Omega_i x + \Phi_i)$. However, here we aim to represent STAF using an architecture that only employs sine activation functions (SIREN). For this purpose, we introduce the following theorem, which holds true for every parametric activation function:

**Theorem 4.2.** *Let $L \geq 2$ and $1 \leq l \leq L$. Consider a neural network as defined in equation 1 with $L$ layers. In addition, let $\boldsymbol{\Omega} = [\Omega_1, ..., \Omega_\tau]^{tr}$, $\boldsymbol{\Phi} = [\Phi_1, ..., \Phi_\tau]^{tr}$, and $\boldsymbol{C} = [C_1, ..., C_\tau]^{tr}$. If the trainable activation function is $\rho^*(x) = \sum_{m=1}^{\tau} \boldsymbol{C}_m \rho(\boldsymbol{\Omega}_m x + \boldsymbol{\Phi}_m)$, then an equivalent neural network with activation function $\rho(x)$ and $L + 1$ layers can be constructed as follows (parameters of the equivalent network are denoted with an overline):*

$$\overline{\boldsymbol{z}^{(0)}} = \gamma(\boldsymbol{r}),$$
$$\overline{\boldsymbol{z}^{(l)}} = \rho\left(\overline{\boldsymbol{W}^{(l)}}\,\overline{\boldsymbol{z}^{(l-1)}} + \overline{\boldsymbol{B}^{(l)}}\right), \quad l = 1, ..., L, \tag{4}$$
$$\overline{f_{\overline{\theta}}}(\boldsymbol{r}) = \overline{\boldsymbol{W}^{(L+1)}}\,\overline{\boldsymbol{z}^{(L)}};$$

*where*

$$\overline{\boldsymbol{W}^{(l)}} = \begin{cases} \boldsymbol{\Omega} \otimes \boldsymbol{W}^{(l)}, & \text{if } l = 1, \\ (\boldsymbol{\Omega} \otimes \boldsymbol{C}^{tr}) \otimes \boldsymbol{W}^{(l)}, & \text{if } l \text{ is even}, \\ (\boldsymbol{\Omega} \otimes \boldsymbol{W}^{(l)}) (\boldsymbol{C}^{tr} \otimes \boldsymbol{I}_{F_{l-1}}), & \text{if } l \text{ is odd}, l > 1, \text{ and } l \neq L + 1, \\ \boldsymbol{C}^{tr} \otimes \boldsymbol{I}_{F_{l-1}}, & \text{if } l \text{ is odd}, l > 1, \text{ and } l = L + 1. \end{cases} \tag{5}$$

*and*

$$\overline{\boldsymbol{B}^{(l)}} = \boldsymbol{\Phi} \otimes \boldsymbol{J}_{F_l}. \tag{6}$$

*in which $\boldsymbol{J}_{F_l}$ is an all-ones $F_l \times 1$ vector. Furthermore, if $L$ is even, then $\overline{f}_{\overline{\theta}}(\boldsymbol{r}) = f_\theta(\boldsymbol{r})$ (we call these networks 'Kronecker equivalent' in this sense).*

The proof of this theorem is provided in the Appendix D.3. As we observed, although a network with the activation function $\rho^*$ can be represented using the activation function $\rho$, it features a unique architecture. These networks are not merely typical MLPs with the activation function $\rho$, as the weights in the Kronecker equivalent network exhibit dependencies due to the Kronecker product.

It is desirable that Theorem equation 4.2 does not depend on the parity of $L$. To achieve this, consider the following remark:

**Remark 5.2.** We can introduce a dummy layer with the activation function $\rho^*$. Specifically, we define $\boldsymbol{z}^{(L)} = \rho^* (f_\theta(\boldsymbol{r}))$, and $\tilde{f}_\theta(\boldsymbol{r}) = \boldsymbol{W}^{(L+1)} \boldsymbol{z}^{(L)} + \boldsymbol{B}^{(L+1)}$, where $\boldsymbol{W}^{(L+1)} = \boldsymbol{O}$. To ensure that $\tilde{f}_\theta(\boldsymbol{r}) = f_\theta(\boldsymbol{r})$, we set $\boldsymbol{B}^{(L+1)} = f_\theta(\boldsymbol{r})$. This approach allows us to construct an equivalent neural network with one more layer.

As a result of Remark 5.2, the equivalent network of a network with a trainable activation function, has either one more layer, or the same number of layers. As an immediate result of Theorem equation 4.2, if we denote the embedding of the first layer of the SIREN equivalent of our network by $\overline{\boldsymbol{\Psi}}$, then

$$\overline{\boldsymbol{\Psi}} = \overline{\boldsymbol{W}^{(1)}} = \boldsymbol{\Omega} \otimes \boldsymbol{W}^{(1)} \in \mathbb{R}^{\tau F_1 \times F_0} \tag{7}$$

which is $\tau$ times bigger than the embedding of the first layer of a SIREN with $\boldsymbol{W}^{(1)} \in \mathbb{R}^{F_1 \times F_0}$. To understand the impact of this increase on expressive power, it suffices to substitute $T$ with $\tau T$ in Theorem equation 4.1. The next theorem will reveal how this change will affect the cardinality of the set of potential frequencies.

**Theorem 4.3.** *(Page 4 of (Kiselman, 2012)) Let $V(T, K) = \left\{ (s_1, s_2, \ldots, s_T) \in \mathbb{Z}^T \mid \sum_{t=1}^{T} |s_t| \leq K \right\}$.[2] Then we have*

$$|V(T, K)| = \sum_{i=0}^{min(K,T)} \binom{i}{K} \binom{i}{T} 2^i \tag{8}$$

*This number is called the Delannoy number. Moreover, for fixed $K$,*

$$|V(T, K)| \sim A_K (2T)^K, \quad T \to +\infty. \tag{9}$$

As an immediate result of this theorem, for large values of $T$, we have $\frac{|V(\tau T, K)|}{|V(T,K)|} \sim \tau^K$. (See Appendix F for an alternative proof.) Now, it is time to analyze the cardinality of the set of potential frequencies:

$$\tilde{\mathcal{H}}(\boldsymbol{\Psi}) = \left\{ \sum_{t=1}^{T} s_t \boldsymbol{\Psi}_t \,\middle|\, (s_1, s_2, \ldots, s_T) \in V(T, J^{L-1}) \right\} \tag{10}$$

or equivalently,

$$\tilde{\mathcal{H}}(\boldsymbol{\Psi}) = \left\{ \boldsymbol{\Psi}^{tr} [s_1, ..., s_T]^{tr} \,\middle|\, s_t \in \mathbb{Z} \wedge \sum_{t=1}^{T} |s_t| \leq J^{L-1} \right\} \tag{11}$$

The cardinality of the set $\tilde{\mathcal{H}}(\boldsymbol{\Psi})$ is bounded above by $V(T, J^{L-1})$. If $\boldsymbol{\Psi}^{tr}$, is injective on the integer lattice $\mathbb{Z}^T$, then $|\tilde{\mathcal{H}}(\boldsymbol{\Psi})| = |V(T, J^{L-1})|$. However, in general, analyzing how a linear transformation

---

[2] We use $V$ to denote these points as cells in a $T$-dimensional von Neumann neighborhood of $K$, clarifying that $V$ does not represent a vector space.

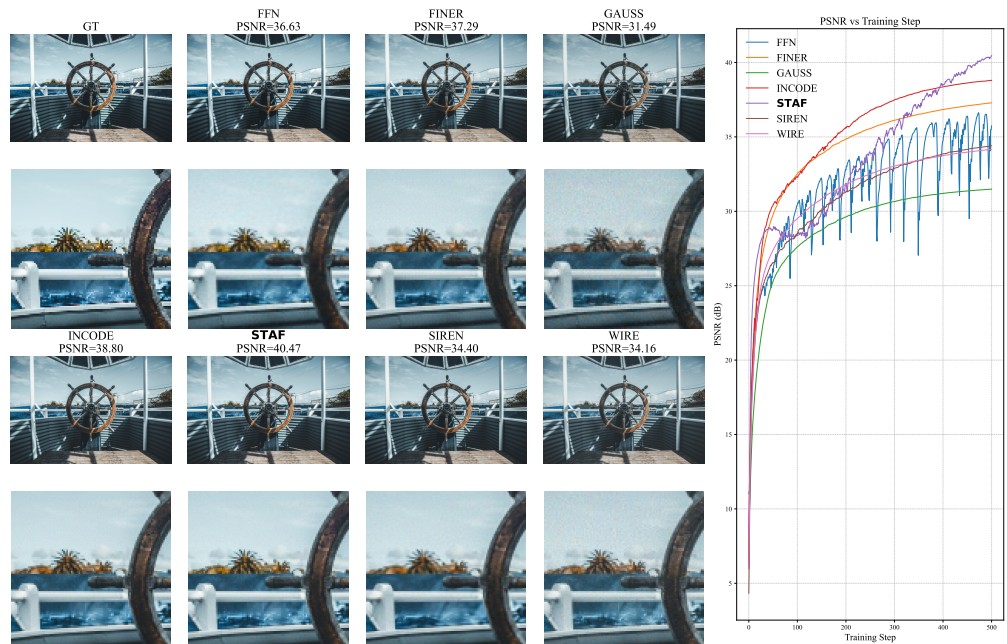

Figure 3: (Left) Visualization of image representations with **STAF** vs. other activations. (Right) PSNR over 500 iterations. Image from DIV2K (Timofte et al., 2018), downsampled $1/4$ to $510 \times 339$.

affects the size of a convex body can be approached using the geometry of numbers (Matousek, 2013) or additive geometry (Tao & Vu, 2006). To simplify the analysis and preserve the size of $\widetilde{\mathcal{H}}(\boldsymbol{\Psi})$ as large as possible, we can slightly perturb the matrix $\boldsymbol{\Psi}^{tr}$ such that its kernel contains no points with rational coordinates, except the origin. This is a much stronger condition than having no integer lattice points in the kernel. To address this, we introduce a lemma. It's worth noting that we can assume the matrices are stored with rational entries, as they are typically represented in computers using floating-point numbers. In our subsequent analysis, however, assuming rational entries for just one column of the matrix $\boldsymbol{\Psi}$ is sufficient.

**Lemma 4.4.** *Let $\boldsymbol{A} \in \mathbb{R}^{D \times T}$, and for one of its rows, like $r$'th row, we have $\boldsymbol{A}_r \in \mathbb{Q}^T$. Then, in every neighborhood of $\boldsymbol{A}$, there is a matrix $\hat{\boldsymbol{A}}$ such that $Ker(\hat{\boldsymbol{A}}) \cap \mathbb{Q}^T = \boldsymbol{O}$.*

(The proof is provided in the Appendix D.4.) Consider Lemma equation 4.4, where we let $\boldsymbol{A} = \boldsymbol{\Psi}^{tr}$. Thus, for every neighborhood of $\boldsymbol{\Psi}^{tr}$, there exists a matrix $\hat{\boldsymbol{\Psi}}^{tr}$ such that $Ker(\hat{\boldsymbol{\Psi}}^{tr}) \cap \mathbb{Q}^T = \boldsymbol{O}$; in other words, $\hat{\boldsymbol{\Psi}}^{tr}$ is injective over rational points, and consequently over integer lattice points. This guarantees that $|\widetilde{\mathcal{H}}(\hat{\boldsymbol{\Psi}})| = |V(T, J^{L-1})|$. Therefore, this section demonstrated that, in comparison to SIREN, STAF can substantially increase the size of the set of potential frequencies by a factor of $\tau^K$, highlighting how the Kronecker product boosts the activation function's expressiveness.

## 5 EXPERIMENTAL RESULTS

We evaluated SOTA models for image, audio, and shape representations, inverse problems such as super-resolution and image denoising, and NeRFs. Specifically, we used an MLP architecture with 3 hidden layers and 256 hidden nodes. The models tested included INCODE, FINER, WIRE, Gauss, FFN, SIREN, ReLU with positional encoding (PEMLP), and MFN (Kazerouni et al., 2024; Liu et al., 2024a; Saragadam et al., 2023; Ramasinghe & Lucey, 2022; Sitzmann et al., 2020; Tancik et al., 2020; Fathony et al., 2020). All experiments used NVIDIA RTX 3090 or A40 GPUs. Our implementation builds on SIREN, WIRE, and INCODE codebases. Learning rates followed optimal settings from original works, and all models were trained with Adam for consistency. STAF was initialized as described in Section 3.5, while other models followed their original initialization strategies. We used $\tau = 5$ for all tasks, except image denoising ($\tau = 2$). We also included a detailed NTK analysis of our model, interesting ablation studies, and more implementation details in *Appendix*.

### 5.1 SIGNAL REPRESENTATIONS

**Image:** We evaluated image representation across multiple datasets and resolutions: DIV2K (Timofte et al., 2018) ($\sim 510 \times 340$, Figures 3 and 5 and Table 6), KODAK (Mehta, 2022; Franzen, 1999)

Table 1: IoU (↑) and Chamfer (CD) (↓) across 3D shapes. best and second-best performance.

| Method | Armadillo | | Dragon | | Lucy | | Thai | | Avg. | |
|---|---|---|---|---|---|---|---|---|---|---|
| | IoU | CD | IoU | CD | IoU | CD | IoU | CD | IoU | CD |
| PEMLP | 0.9958 | 3.70e-7 | 0.9966 | 2.73e-7 | 0.9920 | 1.80e-6 | 0.9911 | 2.05e-6 | 0.9939 | 1.12e-6 |
| SIREN | 0.9962 | 3.62e-7 | 0.9971 | 2.60e-7 | 0.9892 | 2.19e-6 | 0.9929 | 9.59e-7 | 0.9939 | 9.43e-7 |
| WIRE | 0.9721 | 6.54e-6 | 0.9749 | 4.46e-6 | 0.9554 | 2.06e-5 | 0.9507 | 1.55e-5 | 0.9633 | 1.18e-5 |
| FINER | 0.9965 | 3.57e-7 | 0.9958 | 3.06e-7 | 0.9962 | 1.49e-6 | 0.9923 | 1.15e-6 | 0.9952 | 8.24e-7 |
| INCODE | 0.9964 | 3.54e-7 | 0.9969 | 2.65e-7 | 0.9946 | 1.60e-6 | 0.9924 | 1.42e-6 | 0.9951 | 9.10e-7 |
| STAF | 0.9972 | 3.51e-7 | 0.9973 | 2.55e-7 | 0.9971 | 1.39e-6 | 0.9935 | 9.20e-7 | 0.9963 | 7.29e-7 |

Table 2: PSNR (↑) and LPIPS (↓) across different NeRF objects. best and second-best performance.

| Method | LEGO | | Drums | | Chair | | Hotdog | | Ship | |
|---|---|---|---|---|---|---|---|---|---|---|
| | PSNR | LPIPS | PSNR | LPIPS | PSNR | LPIPS | PSNR | LPIPS | PSNR | LPIPS |
| PEMLP | 25.96 | 0.127 | 22.26 | 0.147 | 28.49 | 0.084 | 31.96 | 0.053 | 25.65 | 0.217 |
| Gauss | 25.15 | 0.143 | 22.05 | 0.167 | 28.87 | 0.087 | 32.39 | 0.056 | 25.07 | 0.222 |
| SIREN | 26.27 | 0.159 | 22.94 | 0.168 | 29.71 | 0.087 | 32.85 | 0.058 | 26.00 | 0.220 |
| WIRE | 25.31 | 0.150 | 21.89 | 0.165 | 28.63 | 0.088 | 32.14 | 0.061 | 25.77 | 0.225 |
| FINER | 26.62 | 0.152 | 23.21 | 0.175 | 29.93 | 0.087 | 33.64 | 0.058 | 26.20 | 0.229 |
| STAF | 26.74 | 0.107 | 23.24 | 0.156 | 30.17 | 0.084 | 33.29 | 0.058 | 26.28 | 0.206 |

(24 images, $256 \times 256$, Appendix B.3), CelebA (Liu et al., 2015) (19,867 images, $128 \times 128$, Appendix B.3), and the high-resolution Tokyo panorama (Dobson, 2018) ($6144 \times 2324$, Appendix B.2).

Figure 3 (left) shows that STAF produces sharper edges and more faithful texture recovery than the baselines, surpassing the second-best method, INCODE, by +1.67 dB. The right panel quantifies reconstruction quality through PSNR over 500 training steps, showing STAF's effectiveness in addressing the capacity-convergence gap in INR models. Figure 2 shows activation maps learned during the image reconstruction task. STAF produces more detailed and higher-quality reconstructions compared to SIREN and WIRE, highlighting its ability to capture complex features more effectively.

**Shape:** The quantitative and qualitative results of the shape representation are shown in Table 1 and Figure 7. Using the Stanford 3D Scanning Repository (Laboratory, 1999) and following the INODE strategy (Kazerouni et al., 2024), we generated an occupancy volume by sampling points on a grid $512 \times 512 \times 512$, assigning 1 to voxels inside the object and 0 outside. The results demonstrate STAF's capability to effectively capture both fine and coarse 3D shape details, achieving higher Intersection over Union (IoU) and lower Chamfer distance (CD) than other methods.

**Audio:** For the audio task, we used a 7-second clip from Bach's Cello Suite No. 1: Prelude (Sitzmann et al., 2020), sampled at 44,100 Hz. Figure 6 illustrates the waveforms and reconstruction errors, where STAF demonstrates the highest PSNR, the lowest reconstruction error, and superior fidelity.

## 5.2 INVERSE PROBLEMS

The results in Figures 8 and 9 show that **STAF** consistently outperforms other activation-based implicit models in both super-resolution and denoising. As interpolants, INRs carry inherent biases that can be exploited for inverse problems such as super-resolution. In $4\times$ super-resolution, STAF achieves the best performance (30.54 dB PSNR, 0.89 SSIM), surpassing INCODE (29.88 dB) and FFN (29.41 dB). While SIREN and FINER recover textures reasonably well, they fail to capture high-frequency details, and Gauss suffers from heavy blurring. For denoising, we simulated severe low-light conditions by adding Poisson-distributed photon noise (mean count = 10), producing highly corrupted images. STAF again achieves the highest PSNR (24.19 dB), effectively suppressing noise and retaining details, whereas FINER and FFN show color artifacts. These results demonstrate STAF's effectiveness in high-resolution recovery and inverse problem settings.

## 5.3 NEURAL RADIANCE FIELDS (NERFS)

NeRFs (Mildenhall et al., 2020) utilize INRs and volumetric rendering, where MLPs with ReLU activations and positional encoding are trained to model scenes for novel view synthesis. These models learn a continuous function over 3D space $(x, y, z)$ and view directions $(\theta, \phi)$ to predict color and density at each location. This setup enables the reconstruction of new views by simulating how rays travel through the scene from different camera viewpoints. We evaluate the use of STAF within the NeRF framework, deliberately omitting positional encodings. Our results, shown in Table 2, indicate that STAF achieves superior performance. Additional visual results and training details are provided in Appendix B.1.

## 6 CONCLUSION

We presented a unified view of trainable sinusoidal activations for INRs and instantiated it with STAF, a Fourier-series activation whose parameters are learned. Our theory explains increased frequency capacity and improved optimization via NTK analysis and a Kronecker-equivalence construction; our initialization ensures well-scaled post-activations. Empirically, STAF is competitive and often superior across several INR tasks. Periodic activations do not remove spectral bias outright, but they improve the practical capacity–convergence trade-off, offering a simple, general recipe for higher-fidelity INRs.

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

CONTENTS

APPENDIX

# A  NEURAL TANGENT KERNEL

The Neural Tangent Kernel (NTK) is a significant concept in the theoretical understanding of neural networks, particularly in the context of their training dynamics (Jacot et al., 2018). To be self-contained, we provide an explanation of the NTK and its background in kernel methods. We believe this will be beneficial for readers, as previous papers on implicit neural representation using the NTK concept have not adequately explained the NTK or the significance of its eigenvalues and eigenfunctions.

A kernel is a function $K(\mathbf{x}, \tilde{\mathbf{x}})$ used in integral transforms to define an operator that maps a function $f$ to another function $T_f$ through the integral equation

$$T_f(\mathbf{x}) = \int K(\mathbf{x}, \tilde{\mathbf{x}})f(\tilde{\mathbf{x}})\, d\tilde{\mathbf{x}}.$$

Since $T_f$ is a linear operator with respect to $f$, we can discuss its eigenvalues and eigenfunctions. The eigenvalues and eigenfunctions of a kernel are the scalar values $\lambda$ and the corresponding functions $\zeta(\mathbf{x})$ that satisfy the following equation (Ghojogh et al., 2021)

$$\int K(\mathbf{x}, \tilde{\mathbf{x}})\zeta(\tilde{\mathbf{x}})\, d\tilde{\mathbf{x}} = \lambda\zeta(\mathbf{x}).$$

In the context of neural networks, the concept of a kernel becomes particularly remarkable when analyzing the network's behavior in the infinite-width limit. Kernels in machine learning, such as the Radial Basis Function (RBF) kernel or polynomial kernel, are used to measure similarity between data points in a high-dimensional feature space. These kernels allow the application of linear methods to non-linear problems by implicitly mapping the input data into a higher-dimensional space (Braun, 2005).

The NTK extends this idea by considering the evolution of a neural network's outputs during training. When a neural network is infinitely wide, its behavior can be closely approximated by a kernel method. In this case, the kernel in question is the NTK, which emerges from the first-order Taylor series approximation (or tangent plane approximation) of the network's outputs.

Formally, for a neural network $f(\mathbf{x}; \boldsymbol{\theta})$ with input $\mathbf{x}$ and parameters $\boldsymbol{\theta}$, the NTK, denoted as $K^{(L)}(\mathbf{x}, \tilde{\mathbf{x}})$, is defined as:

$$K^{(L)}(\mathbf{x}, \tilde{\mathbf{x}}) = \langle \nabla_{\boldsymbol{\theta}} f(\mathbf{x}; \boldsymbol{\theta}), \nabla_{\boldsymbol{\theta}} f(\tilde{\mathbf{x}}; \boldsymbol{\theta}) \rangle,$$

where $\nabla_{\boldsymbol{\theta}} f(\mathbf{x}; \boldsymbol{\theta})$ represents the gradient of the network output with respect to its parameters.

There are two methods for calculating the NTK: the analytic approach and the empirical approach (Novak et al., 2019; Chen et al., 2022). In the paper, we derived the analytic NTK of a neural network that uses our activation function, as detailed in the appendix. However, for our experimental purposes, we utilized the empirical NTK. It is worth noting that calculating the NTK for real-world networks is highly challenging, and typically not computationally possible (Mohamadi et al., 2023).

Just like the computation of NTK, there are analytic and empirical methods for calculating the eigenvalues and eigenfunctions of a kernel (Williams & Seeger, 2000). These values play a crucial role in characterizing neural network training. For instance, it has been shown that the eigenvalues of the NTK determine the convergence rate (Wang et al., 2022; Bai et al., 2023). Specifically, components of the target function associated with kernel eigenvectors having larger eigenvalues are learned faster (Wang et al., 2022; Tancik et al., 2020). In fully-connected networks, the eigenvectors corresponding to higher eigenvalues of the NTK matrix generally represent lower frequency components (Wang et al., 2022). Furthermore, the eigenfunctions of an NTK can illustrate how effectively a model learns a signal dictionary (Yüce et al., 2022).

Figure 4(a) illustrates the eigenfunctions of various NTKs using different activation functions. As shown, the STAF activation function results in finer eigenfunctions, which intuitively enhances the ability to learn and reconstruct higher frequency components. Additionally, Figure 4(b) presents the eigenvalues of different NTKs with various activation functions. The results indicate that STAF

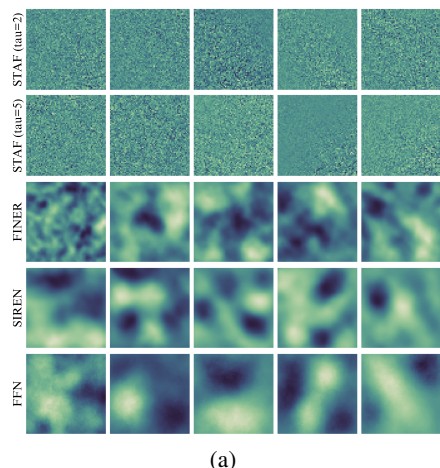
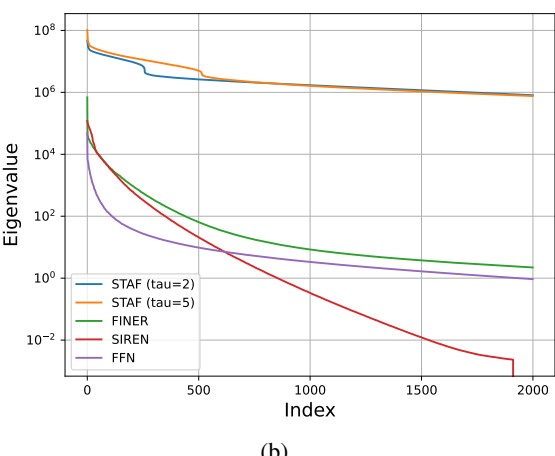

(a)               (b)

Figure 4: (a) The first five eigenfunctions of the empirical NTK of STAF ($\tau = 2, 5$), FINER, SIREN, and FFN. (b) The eigenvalue spectrum of the empirical NTK of the same models. Interestingly, the eigenvalue spectrum and the first five eigenfunctions for different $\tau$'s look very similar to each other, which shows that we can achieve on-par results even with fewer $\tau$ (or number of parameters), which is consistent with our result in Table 5. Also note that SIREN is similar to STAF with $\tau = 1$, except for the initialization scheme, as well as the use of trainable frequencies and phases. However, these differences result in a significant performance gap.

produces higher eigenvalues, leading to a faster convergence rate during training. Moreover, STAF also generates a greater number of eigenvalues, compared to ReLU and SIREN. Having more eigenvalues is beneficial because it suggests a richer and more expressive kernel, capable of capturing a wider range of features and details in the data.

## A.1 ANALYTIC NTK

In this section, we compute the analytic NTK for a neural network that uses the proposed activation function (STAF), following the notation from (Radhakrishnan, 2024). Interested readers can also refer to (Jacot et al., 2018) and (Golikov et al., 2022). However, we chose (Radhakrishnan, 2024) for its clarity and ease of understanding. According to (Radhakrishnan, 2024), the NTK of an activation function for a neural network with $L - 1$ hidden layers is as follows.

**Theorem A.1.** *(Theorem 1 of (Radhakrishnan, 2024), Lecture 6) For $\boldsymbol{x} \in \mathcal{S}^{d-1}$, let $f_{\boldsymbol{x}}^{(L)}(\boldsymbol{w}) : \mathbb{R}^p \to \mathbb{R}$ denote a neural network with $L - 1$ hidden layers such that:*

$$f_{\boldsymbol{x}}^{(L)}(\boldsymbol{w}) = \boldsymbol{W^{(L)}} \frac{1}{\sqrt{F_{L-1}}} \phi \left( \boldsymbol{W^{(L-1)}} \frac{1}{\sqrt{F_{L-2}}} \phi \left( \dots \boldsymbol{W^{(2)}} \frac{1}{\sqrt{F_1}} \phi \left( \boldsymbol{W^{(1)}} \boldsymbol{x} \right) \dots \right) \right); \quad (12)$$

*where $W^{(i)} \in \mathbb{R}^{F_i \times F_{i-1}}$ for $i \in \{1, \dots, L\}$ with $F_0 = d$, $F_L = 1$, and $\phi : \mathbb{R} \to \mathbb{R}$ is an element-wise activation function. As $F_1, F_2, \dots, F_{L-1} \to \infty$ in order, the Neural Network Gaussian Process (NNGP), denoted as $\Sigma^{(L)}$, and the NTK, denoted as $K^{(L)}$, of $f_{\boldsymbol{x}}(\boldsymbol{w})$ are given by:*

$$\Sigma^{(L)}(\boldsymbol{x}, \tilde{\boldsymbol{x}}) = \check{\phi} \left( \Sigma^{(L-1)}(\boldsymbol{x}, \tilde{\boldsymbol{x}}) \right); \quad \Sigma^{(0)}(\boldsymbol{x}, \tilde{\boldsymbol{x}}) = \boldsymbol{x}^T \tilde{\boldsymbol{x}}$$

$$K^{(L)}(\boldsymbol{x}, \tilde{\boldsymbol{x}}) = \Sigma^{(L)}(\boldsymbol{x}, \tilde{\boldsymbol{x}}) + K^{(L-1)}(\boldsymbol{x}, \tilde{\boldsymbol{x}}) \check{\phi}' \left( \Sigma^{(L-1)}(\boldsymbol{x}, \tilde{\boldsymbol{x}}) \right); \quad (13)$$

$$K^{(0)}(\boldsymbol{x}, \tilde{\boldsymbol{x}}) = \boldsymbol{x}^T \tilde{\boldsymbol{x}}$$

*where $\check{\phi} : [-1, 1] \to \mathbb{R}$ is the dual activation for $\phi$, and is calculated as follows:*

$$\check{\phi}(\xi) = \mathbb{E}_{(u,v) \sim \mathcal{N}(0,\boldsymbol{\Lambda})}[\phi(u)\phi(v)] \quad where \; \boldsymbol{\Lambda} = \begin{bmatrix} 1 & \xi \\ \xi & 1 \end{bmatrix}. \quad (14)$$

*Furthermore, $\phi$ is normalized such that $\check{\phi}(1) = 1$.*

Consequently, it suffices to calculate $\check{\phi}$. It has been calculated in the following theorem. Just like what mentioned in (Wang et al., 2023), we assume that the optimization of neural networks with STAF can be decomposed into two phases, where we learn the coefficients of STAF in the first phase and then train the parameters of the neural network in the second phase. This assumption is reasonable as the number of parameters of STAF is far less than that of networks, and they quickly converge at the early stage of training. As a result, in the following theorem, all the parameters except weights are fixed, since they have been obtained in the first phase of training.

**Theorem A.2.** *Let $\rho^*$ be the proposed activation function (STAF). Then*

$$\check{\rho^*}(\xi) = \sum_{i=1}^{\tau}\sum_{j=1}^{\tau} C_i C_j \Delta_{i,j}$$

$$= \frac{1}{2}\sum_{i=1}^{\tau}\sum_{j=1}^{\tau} C_i C_j e^{\frac{-1}{2}\left(\Omega_i^2+\Omega_j^2\right)}\left(e^{\Omega_i\Omega_j\xi}\cos(\Phi_i-\Phi_j)+e^{-\Omega_i\Omega_j\xi}\cos(\Phi_i+\Phi_j)\right) \tag{15}$$

*Therefore,*

$$\check{\rho^*}'(\xi) = \frac{1}{2}\sum_{i=1}^{\tau} C_i\Omega_i \sum_{j=1}^{\tau}\left[C_j\Omega_j e^{\frac{-1}{2}\left(\Omega_i^2+\Omega_j^2\right)}\left(e^{\Omega_i\Omega_j\xi}\cos(\Phi_i-\Phi_j)-e^{-\Omega_i\Omega_j\xi}\cos(\Phi_i+\Phi_j)\right)\right]. \tag{16}$$

*Proof.*

$$\check{\rho^*}(\xi) = \mathbb{E}_{(u,v)\sim\mathcal{N}(0,\mathbf{\Lambda})}[\rho^*(u)\rho^*(v)]$$

$$= \mathbb{E}_{(u,v)\sim\mathcal{N}(0,\mathbf{\Lambda})}\left[\sum_{i=1}^{\tau} C_i\sin(\Omega_i u+\Phi_i)\sum_{i=1}^{\tau} C_i\sin(\Omega_i v+\Phi_i)\right]$$

$$= \mathbb{E}_{(u,v)\sim\mathcal{N}(0,\mathbf{\Lambda})}\left[\sum_{i=1}^{\tau}\sum_{j=1}^{\tau} C_i C_j\sin(\Omega_i u+\Phi_i)\sin(\Omega_j v+\Phi_j)\right]$$

$$= \sum_{i=1}^{\tau}\sum_{j=1}^{\tau} C_i C_j\mathbb{E}_{(u,v)\sim\mathcal{N}(0,\mathbf{\Lambda})}\left(\sin(\Omega_i u+\Phi_i)\sin(\Omega_j v+\Phi_j)\right). \tag{17}$$

So, we need to compute the following expectation:

$$\Delta_{i,j} = \mathbb{E}_{(u,v)\sim\mathcal{N}(0,\mathbf{\Lambda})}\left(\sin(\Omega_i u+\Phi_i)\sin(\Omega_j v+\Phi_j)\right) \tag{18}$$

Note that for a random vector $\mathbf{X} = (X_1,\ldots,X_d)^T$ with mean vector $\boldsymbol{\mu}$ and covariance matrix $\mathbf{\Lambda}$, the joint probability density function (PDF) is as follows:

$$f_{\mathbf{X}}(\mathbf{x}) = (2\pi)^{-d/2}\det(\mathbf{\Lambda})^{-1/2}e^{\left(\frac{-1}{2}(\mathbf{x}-\boldsymbol{\mu})^T\mathbf{\Lambda}^{-1}(\mathbf{x}-\boldsymbol{\mu})\right)}. \tag{19}$$

As a result, since $\mathbf{\Lambda}^{-1} = \frac{1}{1-\xi^2}\begin{bmatrix} 1 & -\xi \\ -\xi & 1 \end{bmatrix}$, we will have:

$$f_{U,V}(u,v) = \frac{1}{2\pi\sqrt{1-\xi^2}}e^{-\frac{1}{2}(u\quad v)\mathbf{\Lambda}^{-1}\binom{u}{v}} = \frac{1}{2\pi\sqrt{1-\xi^2}}e^{\frac{-1}{2(1-\xi^2)}(u\quad v)\begin{pmatrix} 1 & -\xi \\ -\xi & 1 \end{pmatrix}\binom{u}{v}}$$

$$= \frac{1}{2\pi\sqrt{1-\xi^2}}e^{\frac{-(u^2-2\xi uv+v^2)}{2(1-\xi^2)}}. \tag{20}$$

Consequently, using Equations (17) and (18), we have

$$\Delta_{i,j} = \int_{-\infty}^{\infty}\int_{-\infty}^{\infty}\left(\sin(\Omega_i u+\Phi_i)\sin(\Omega_j v+\Phi_j)f_{U,V}(u,v)\right)dudv$$

$$= \frac{1}{2\pi\sqrt{1-\xi^2}}\int_{-\infty}^{\infty}\sin(\Omega_j v+\Phi_j)I_1 dv; \tag{21}$$

where

$$I_1 = \int_{-\infty}^{\infty} \sin(\Omega_i u + \Phi_i)e^{\frac{-(u^2 - 2\xi uv + v^2)}{2(1-\xi^2)}}\, du = e^{\frac{-v^2}{2(1-\xi^2)}} \int_{-\infty}^{\infty} \sin(\Omega_i u + \Phi_i)e^{\frac{-(u^2 - 2\xi uv)}{2(1-\xi^2)}}\, du$$

$$= e^{\frac{-v^2 + \xi^2 v^2}{2(1-\xi^2)}} \int_{-\infty}^{\infty} \sin(\Omega_i u + \Phi_i)e^{\frac{-(u^2 - 2\xi uv + \xi^2 v^2)}{2(1-\xi^2)}}\, du = e^{-v^2/2} \int_{-\infty}^{\infty} \sin(\Omega_i u + \Phi_i)e^{\frac{-(u-\xi v)^2}{2(1-\xi^2)}}\, du \tag{22}$$

By assuming $\eta = u - \xi v$ we will have:

$$I_1 = e^{-v^2/2} \int_{-\infty}^{\infty} \sin(\Omega_i(\eta + \xi v) + \Phi_i)e^{\frac{-\eta^2}{2(1-\xi^2)}}\, d\eta \tag{23}$$

Before going further, we need to consider the following lemma.

**Lemma A.3.**

$$\int_{-\infty}^{\infty} \cos(\alpha u + \beta)e^{-\gamma u^2}\, du = \sqrt{\frac{\pi}{\gamma}}e^{-\frac{\alpha^2}{4\gamma}}\cos\beta, \tag{24}$$

$$\int_{-\infty}^{\infty} \sin(\alpha u + \beta)e^{-\gamma u^2}\, du = \sqrt{\frac{\pi}{\gamma}}e^{-\frac{\alpha^2}{4\gamma}}\sin\beta \tag{25}$$

*The proof is provided in equation A.1.1.*

Let $\alpha = \Omega_i$, $\beta = \Omega_i \xi v + \Phi_i$, and $\gamma = \frac{1}{2(1-\xi^2)}$. As a result of equation equation 25, we have

$$I_1 = e^{-v^2/2}\sqrt{2\pi(1-\xi^2)}e^{\frac{-\Omega_i^2}{2/(1-\xi^2)}}\sin(\Omega_i \xi v + \Phi_i)$$

$$= \sqrt{2\pi(1-\xi^2)}e^{\frac{-(v^2 + \Omega_i^2(1-\xi^2))}{2}}\sin(\Omega_i \xi v + \Phi_i) \tag{26}$$

Therefore, based on equation 21, we will have

$$\Delta_{i,j} = \frac{1}{2\pi\sqrt{1-\xi^2}} \int_{-\infty}^{\infty} \left[\sin(\Omega_j v + \Phi_j)\sqrt{2\pi(1-\xi^2)}e^{\frac{-(v^2 + \Omega_i^2(1-\xi^2))}{2}}\sin(\Omega_i \xi v + \Phi_i)\right] dv$$

$$= \frac{e^{\frac{(-\Omega_i^2(1-\xi^2))}{2}}}{\sqrt{2\pi}} \int_{-\infty}^{\infty} \left[\sin(\Omega_j v + \Phi_j)e^{-v^2/2}\sin(\Omega_i \xi v + \Phi_i)\right] dv$$

$$= \frac{e^{-\Omega_i^2(1-\xi^2)/2}}{\sqrt{2\pi}} \int_{-\infty}^{\infty} e^{-v^2/2}\,\aleph\, dv \tag{27}$$

where

$$\aleph = \frac{1}{2}\left[\cos(v(\Omega_i \xi - \Omega_j) + \Phi_i - \Phi_j) - \cos(v(\Omega_i \xi + \Omega_j) + \Phi_i + \Phi_j)\right] \tag{28}$$

Therefore,

$$\Delta_{i,j} = \frac{e^{-\Omega_i^2(1-\xi^2)/2}}{2\sqrt{2\pi}} \left(\sqrt{2\pi}e^{-(\Omega_i \xi - \Omega_j)^2/2}\cos(\Phi_i - \Phi_j) + \sqrt{2\pi}e^{-(\Omega_i \xi + \Omega_j)^2/2}\cos(\Phi_i + \Phi_j)\right)$$

$$= \frac{e^{-\Omega_i^2(1-\xi^2)/2}}{2} \left(e^{-(\Omega_i \xi - \Omega_j)^2/2}\cos(\Phi_i - \Phi_j) + e^{-(\Omega_i \xi + \Omega_j)^2/2}\cos(\Phi_i + \Phi_j)\right)$$

$$= \frac{e^{\frac{-\Omega_i^2(1-\xi^2)}{2}}e^{\frac{-(\Omega_i^2 \xi^2 + \Omega_j^2)}{2}}}{2} \left(e^{\Omega_i \Omega_j \xi}\cos(\Phi_i - \Phi_j) + e^{-\Omega_i \Omega_j \xi}\cos(\Phi_i + \Phi_j)\right)$$

$$= \frac{e^{\frac{-1}{2}(\Omega_i^2 + \Omega_j^2)}}{2} \left(e^{\Omega_i \Omega_j \xi}\cos(\Phi_i - \Phi_j) + e^{-\Omega_i \Omega_j \xi}\cos(\Phi_i + \Phi_j)\right) \tag{29}$$

As a result of Equations (17) and (29), we have

$$\check{\rho}^*(\xi) = \sum_{i=1}^{\tau}\sum_{j=1}^{\tau} C_i C_j \Delta_{i,j}$$

$$= \frac{1}{2}\sum_{i=1}^{\tau}\sum_{j=1}^{\tau} C_i C_j e^{\frac{-1}{2}(\Omega_i^2 + \Omega_j^2)}\left(e^{\Omega_i \Omega_j \xi}\cos(\Phi_i - \Phi_j) + e^{-\Omega_i \Omega_j \xi}\cos(\Phi_i + \Phi_j)\right) \tag{30}$$

$$\square$$

### A.1.1 Proof of equation A.3

*Proof.* We want to calculate these integrals:

$$I_1 = \int_{-\infty}^{\infty} \cos(\alpha u + \beta) e^{-\gamma u^2} du,$$

$$I_2 = \int_{-\infty}^{\infty} \sin(\alpha u + \beta) e^{-\gamma u^2} du \tag{31}$$

By adding them we will have

$$I_1 + iI_2 = \int_{-\infty}^{\infty} e^{-\gamma u^2} \big(\cos(\alpha u + \beta) + i\sin(\alpha u + \beta)\big) du = \int_{-\infty}^{\infty} e^{i(\alpha u + \beta)} e^{-\gamma u^2} du$$

$$= e^{i\beta} \int_{-\infty}^{\infty} e^{-\gamma(u^2 + \frac{\alpha i}{\gamma} u)} du = e^{i\beta} \int_{-\infty}^{\infty} e^{-\gamma(u^2 + \frac{\alpha i}{\gamma} u - \frac{\alpha^2}{4\gamma^2})} e^{-\frac{\alpha^2}{4\gamma}} du$$

$$= e^{-\frac{\alpha^2}{4\gamma} + i\beta} \int_{-\infty}^{\infty} e^{-\gamma(u^2 + \frac{\alpha i}{\gamma} u - \frac{\alpha^2}{4\gamma^2})} du = e^{-\frac{\alpha^2}{4\gamma} + i\beta} \underbrace{\int_{-\infty}^{\infty} e^{-\gamma\left(u + \frac{\alpha i}{2\gamma}\right)^2} du}_{I_3} \tag{32}$$

where $i$ is the unit imaginary number. Since we know that the integral of an arbitrary Gaussian function is

$$\int_{-\infty}^{\infty} e^{-a(x+b)^2} dx = \sqrt{\frac{\pi}{a}}, \tag{33}$$

we will have $I_3 = \sqrt{\frac{\pi}{\gamma}}$. Therefore,

$$I_1 + iI_2 = \sqrt{\frac{\pi}{\gamma}} e^{-\frac{\alpha^2}{4\gamma} + i\beta} = \sqrt{\frac{\pi}{\gamma}} e^{-\frac{\alpha^2}{4\gamma}} (\cos\beta + i\sin\beta) \tag{34}$$

As a result,

$$I_1 = \sqrt{\frac{\pi}{\gamma}} e^{-\frac{\alpha^2}{4\gamma}} \cos\beta, \quad I_2 = \sqrt{\frac{\pi}{\gamma}} e^{-\frac{\alpha^2}{4\gamma}} \sin\beta. \tag{35}$$

$\square$

## B Additional Experimental Results

### B.1 Neural Radiance Fields (NeRFs)

To implement our NeRF experiments, we adopted an approach inspired by (Saragadam et al., 2023), utilizing the publicly available `torch-ngp` framework (Tang, 2022) for training. Our architecture consists of two separate MLPs: one predicts the volumetric density ($\sigma$), while the other outputs the color values (RGB). Both networks are implemented as 4-layer MLPs with 128 hidden units per layer. The sigma network received only spatial coordinates $(x, y, z)$, while the RGB network also included viewing direction parameters $(\theta, \phi)$. We conducted experiments on five different scenes, using 100 training images, each downsampled to $400 \times 400$ resolution. The generalization of the model was evaluated using 200 additional test views. All training was performed on an NVIDIA A40 GPU with 48 GB of memory. During training, we used the learning rate $3 \times 10^{-4}$ for STAF, $5 \times 10^{-4}$ for FINER following their codebase, and followed the same optimized learning rates using (Kazerouni et al., 2024) for other methods. For sinusoidal-based methods, we used $\omega_0 = 40$ (SIREN, WIRE, STAF), $\omega_0 = 30$ for FINER, and $\sigma_0 = 40$ (WIRE and Gauss). Apart from ReLU, no positional encoding was used for other nonlinearities to isolate their representational capabilities. We illustrate the visual comparisons of Lego in Figure 10.

### B.2 Results on the Tokyo Image

We additionally evaluated STAF on a high-resolution image to demonstrate its scalability and ability to reconstruct fine details. STAF both quantitatively and qualitatively outperforms all baseline

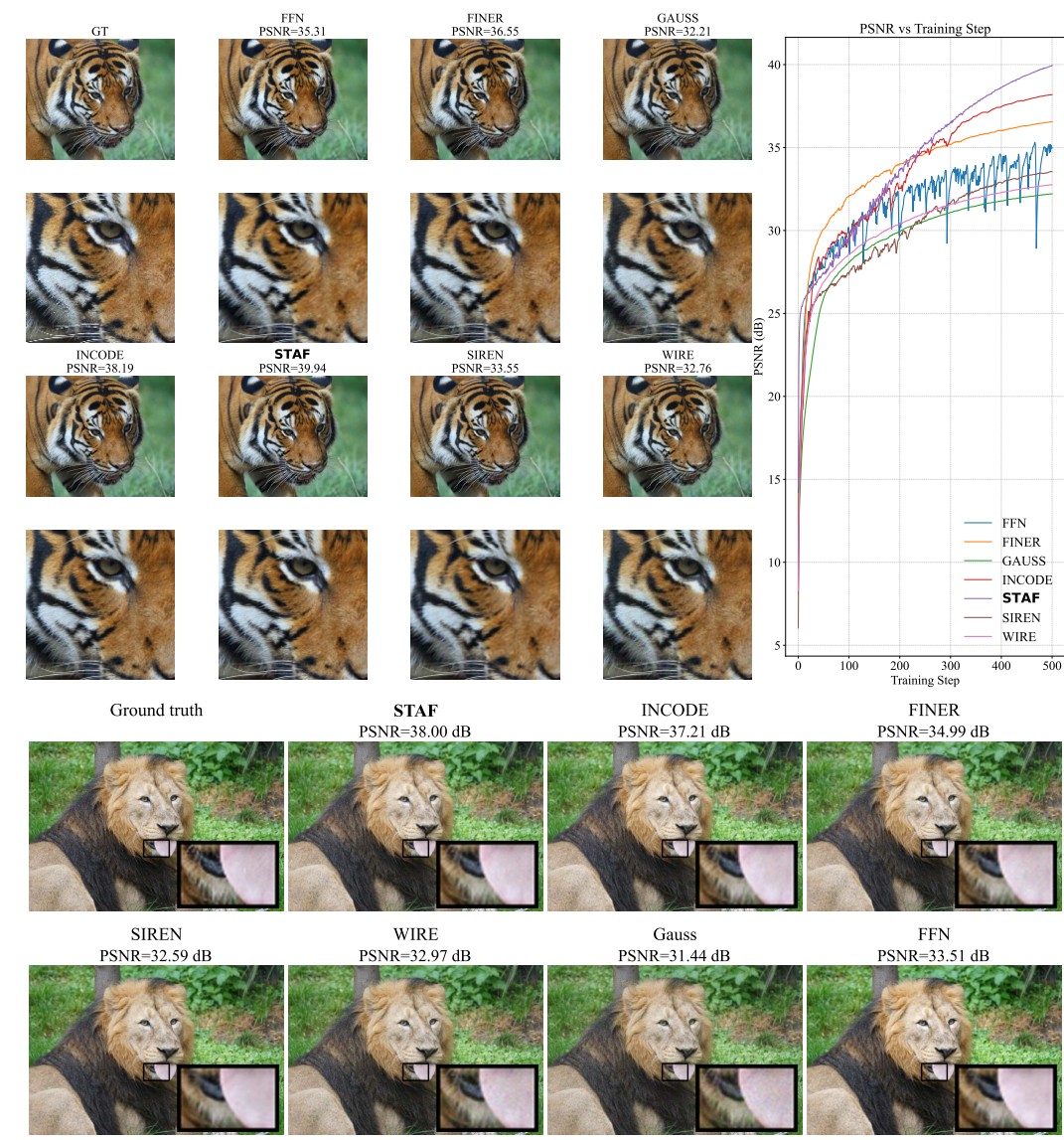

Figure 5: Comparative visualization of image representation with **STAF** and other activation functions.

methods, which is depicted in Figure 11. We have highlighted key differences in Figure 12. As shown, STAF provides a more accurate reconstruction by preserving fine details and colors. For example, in the *first row*, STAF correctly reconstructs the text in green, matching the ground truth, while other methods tend to reconstruct it in blue. Furthermore, in the traffic sign image *(last row)*, only STAF successfully reconstructs the `"No Parking"` sign, faithfully preserving the circular blue background, the red border, and the diagonal slash, closely matching the ground truth. In contrast, InstantNGP (Müller et al., 2022) and SIREN fail to recover the fine structure and distort both color and shape. FINER partially retains the circular shape but loses critical color fidelity. These results demonstrate the superior performance of STAF in capturing fine-grained visual details, which is also supported by the provided error map. In this experiment, we use the PyTorch InstantNGP training pipeline, substituting the original ReLU activations with FINER, STAF, and SIREN.

## B.3 RESULTS ON THE KODAK, CELEBA, AND DIV2K DATASETS

Tables 3 and 4 summarize the quantitative comparisons using PSNR, SSIM, and LPIPS metrics that respectively assess pixel-wise quality, structural similarity, and perceptual quality. On the Kodak

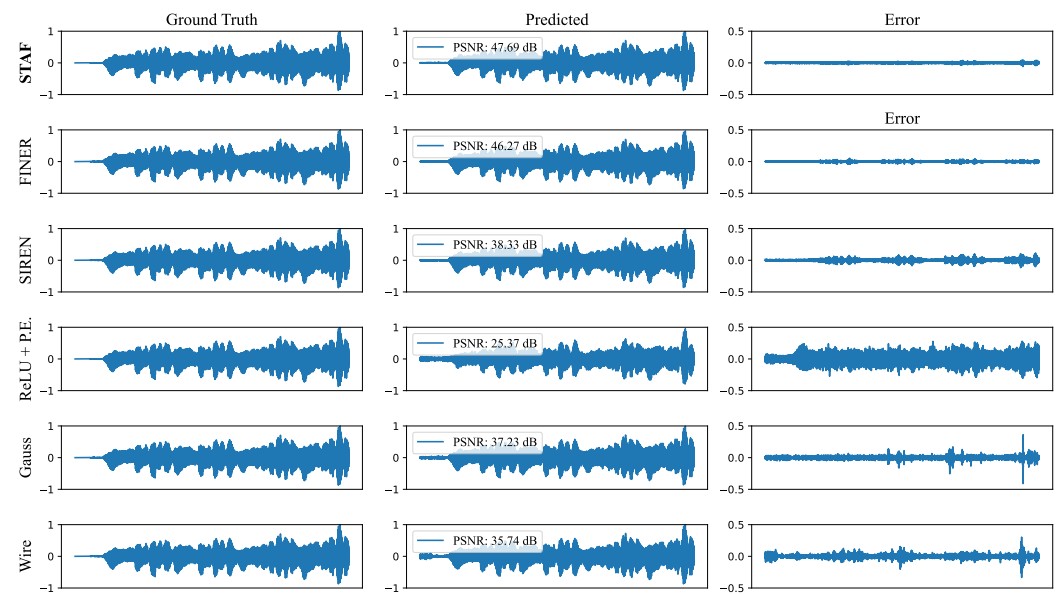

Figure 6: Comparative visualization of audio representation with **STAF** and other activation functions.

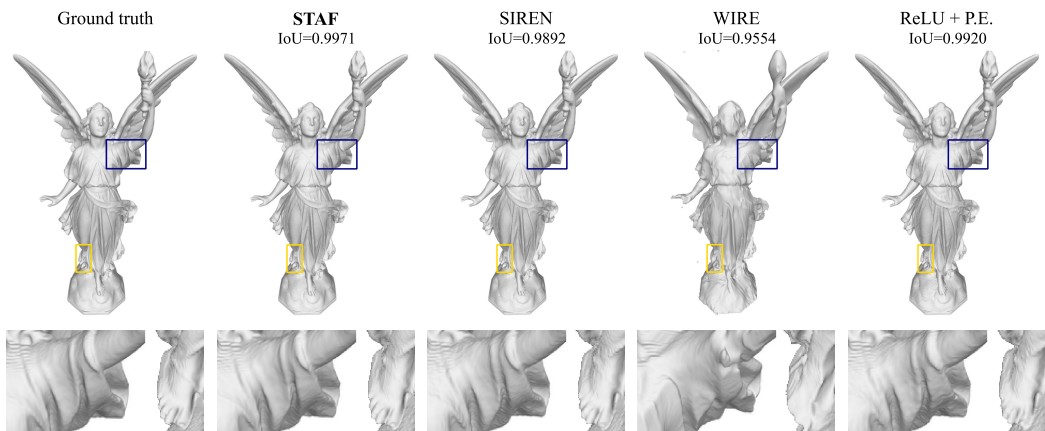

Figure 7: Comparative visualization of shape representation with **STAF** and other activation functions.

dataset (Mehta, 2022; Franzen, 1999), which consists of 24 high-quality $256 \times 256 \times 3$ images, STAF achieves the highest PSNR and SSIM scores, indicating excellent fidelity and structural similarity in the reconstructed images. Similarly, on the CelebA dataset (Liu et al., 2015), containing 19,867 $128 \times 128 \times 3$ images, STAF outperforms other methods, showcasing its ability to generalize across diverse and large-scale datasets. Surprisingly, our results are sensitive to the LPIPS metric, ranking fourth-best on both datasets. This discrepancy stems from the well-known distortion–perception trade-off. STAF minimizes pixel-wise losses (e.g., MSE), which favor smooth reconstructions and lead to high PSNR/SSIM by approximating the conditional mean. However, LPIPS measures perceptual similarity using deep features that are sensitive to high-frequency details and textures, features often averaged out in pixel-accurate reconstructions, resulting in lower perceptual scores. We also report PSNR values on five randomly selected samples from the DIV2K (Timofte et al., 2018) $512 \times 512$ dataset in Table 6, comparing STAF with alternative activation functions. Across all samples, STAF consistently achieves relatively higher reconstruction quality.

## C  ABLATION STUDIES

In this section, we present ablation studies to demonstrate the effectiveness of STAF.

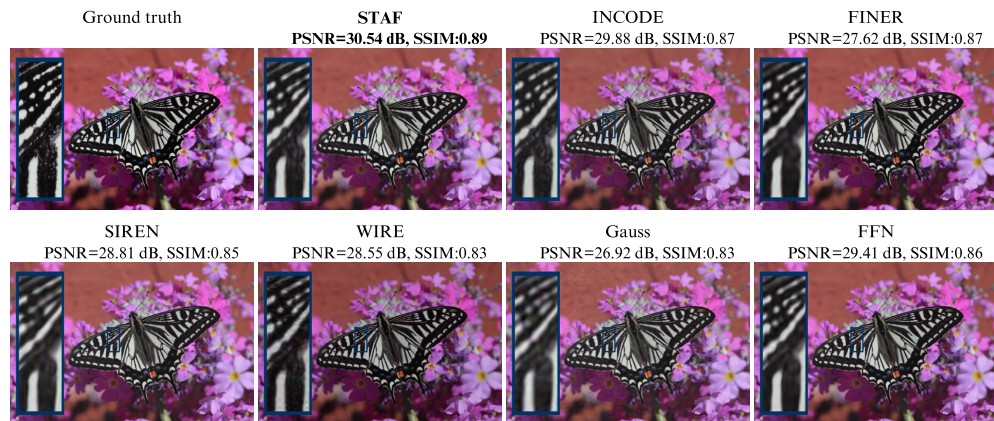

Figure 8: Comparative visualization of 4x super resolution with **STAF** and other activation functions.

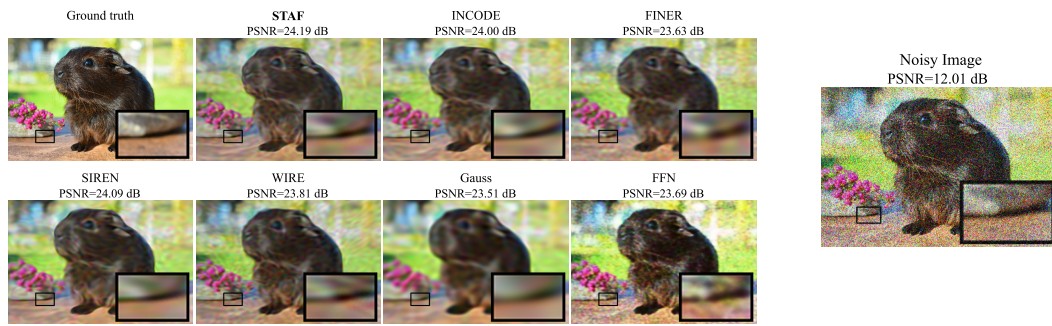

Figure 9: Comparative visualization of image denoising with **STAF** and other activation functions.

### C.1 IMPACT OF AMPLITUDE, FREQUENCY, AND PHASE

Figure 13 illustrates the PSNR (dB) over 500 iterations for different component combinations: **amplitude** ($C_i$'s), **frequency** ($\Omega_i$'s), **phase** ($\Phi_i$'s), and their interactions. The model leveraging all three components (freq + phase + amp) achieves the highest PSNR, significantly outperforming individual and partial combinations. This confirms the importance of integrating amplitude, frequency, and phase in the model design for optimal performance, and validates our initial design choices and mathematical analysis.

Additionally, this graph highlights the varying importance of the parameters in our model. Specifically, the amplitudes exhibit the highest significance, followed by the frequencies, with the phases contributing the least. These findings provide valuable guidance for parameter reduction in scenarios with limited training time or hardware resources, enabling more efficient model optimization.

### C.2 EFFECT OF NUMBER OF SINUSOIDS ($\tau$)

We conducted an ablation study on the number of sinusoids $\tau$ to investigate its impact on model performance and computational efficiency, as summarized in Table 5. The results were obtained on the Kodak dataset (Mehta, 2022), and training and inference times were measured on a single NVIDIA T4 GPU with 16GB of memory. We train models for 500 iterations.

By varying $\tau$, we observed that increasing the number of sinusoids improves reconstruction quality, reflected in higher PSNR and SSIM, and lower LPIPS scores, as well as convergence stability (lower variance). However, these improvements come at the cost of increased training and inference times, and higher model complexity. Therefore, selecting an appropriate $\tau$ involves a trade-off between performance and computational overhead, which can be adjusted depending on application-specific

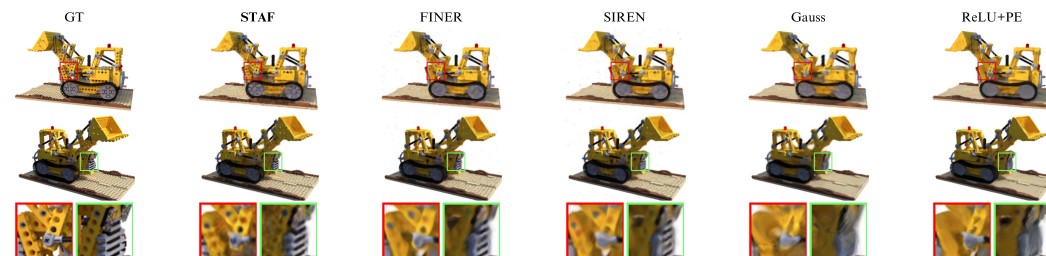

| GT | STAF | FINER | SIREN | Gauss | ReLU+PE |

Figure 10: Comparative visualization of novel view synthesis in NeRF using **STAF** and other activation functions.

Table 3: Kodak dataset (24 images, 1000 training iterations). best and second-best performance.

| Method | PSNR ↑ | SSIM ↑ | LPIPS ↓ |
|---|---|---|---|
| STAF ($\tau = 5$) | $57.13 \pm 9.12$ | $0.993 \pm 0.082$ | $0.011 \pm 0.052$ |
| SIREN | $29.82 \pm 2.31$ | $0.843 \pm 0.054$ | $0.201 \pm 0.081$ |
| FINER | $43.81 \pm 2.45$ | $0.986 \pm 0.010$ | $0.004 \pm 0.004$ |
| WIRE | $29.02 \pm 2.54$ | $0.828 \pm 0.074$ | $0.101 \pm 0.096$ |
| INCODE | $47.97 \pm 1.23$ | $0.992 \pm 1.535$ | $0.001 \pm 0.001$ |
| MFN | $44.61 \pm 1.52$ | $0.986 \pm 0.006$ | $0.001 \pm 0.001$ |
| Gauss | $31.75 \pm 2.34$ | $0.886 \pm 0.026$ | $0.068 \pm 0.030$ |
| ReLU + PE | $28.37 \pm 2.012$ | $0.761 \pm 0.049$ | $0.182 \pm 0.047$ |

Table 4: CelebA dataset (19,867 images, 250 training iterations). best and second-best performance.

| Method | PSNR ↑ | SSIM ↑ | LPIPS ↓ |
|---|---|---|---|
| STAF ($\tau = 5$) | $70.932 \pm 4.248$ | $0.9970 \pm 0.0075$ | $0.0005 \pm 0.0017$ |
| SIREN | $58.251 \pm 2.322$ | $0.9955 \pm 0.0025$ | $0.0006 \pm 0.0004$ |
| FINER | $48.96 \pm 2.136$ | $0.9932 \pm 0.0029$ | $0.0003 \pm 0.0003$ |
| WIRE | $39.283 \pm 2.403$ | $0.9773 \pm 0.0144$ | $0.0082 \pm 0.0052$ |
| MFN | $44.124 \pm 2.912$ | $0.9890 \pm 0.0059$ | $0.0021 \pm 0.0021$ |
| INCODE | $59.853 \pm 5.523$ | $0.9964 \pm 0.0019$ | $0.0001 \pm 0.0003$ |
| GAUSS | $47.547 \pm 2.807$ | $0.9934 \pm 0.0055$ | $0.0003 \pm 0.0004$ |
| ReLU + PE | $31.088 \pm 2.318$ | $0.8899 \pm 0.0372$ | $0.0865 \pm 0.0414$ |

constraints. We set $\tau = 5$ as a good trade-off between quality and training time for all tasks, except for denoising, where a smaller value of $\tau = 2$ yielded better performance.

## C.3 COMPARATIVE ANALYSIS OF ACTIVATION STRATEGIES

Figure 14 aligns with the described strategies in Section 3.4 for implementing STAF's parametric activation functions. The per-neuron activation (green curve) achieves the highest PSNR, demonstrating superior expressiveness, but at the cost of a significant parameter increase, as expected. The network-wide activation (blue curve) shows the weakest performance, reflecting limited expressiveness due to shared activation functions across the entire network. The layer-wise activation (orange curve) offers a balanced trade-off, achieving nearly the same performance as per-neuron activation while requiring far fewer additional parameters (e.g., 225 parameters for a 3-layer MLP with 25 terms). This supports its use as an efficient and effective strategy, as highlighted in Section 3.4.

## C.4 COMPUTATIONAL COST

We provide a detailed analysis in Table 7, comparing all trainable and non-trainable methods in terms of parameter count and FLOPs to better assess the computational cost and complexity of STAF on a single image reconstruction task (on the tiger image in Figure 5). As shown, increasing $\tau$ results in only a minimal rise in parameter count, 60 parameters for $\tau = 5$ and 120 for $\tau = 10$, which is negligible relative to the total number of model parameters. This increase yields improved reconstruction quality without a significant rise in FLOPs. Moreover, it is worth noting that although our STAF models have fewer parameters than INCODE, MFN, FFN, and ReLU+PE, they outperform all of them across all metrics. We used the Calflops package (MrYxJ, 2023) to calculate the FLOP count.

## C.5 INITIALIZATION

Note that while our initialization technique is designed to address the shortcomings of SIREN's, it remains compatible with existing approaches like SIREN, as it operates on the coefficients rather than the weights. As demonstrated in Figure 17, the central limit theorem remains valid for the post-activation values, ensuring that the distribution of dot-product values follows $\mathcal{N}(0, 1)$ across different $\tau$ values. In addition, with increasing $\tau$, STAF exhibits a greater increase in maximum frequency compared to SIREN, allowing it to capture a broader range of frequencies more effectively.

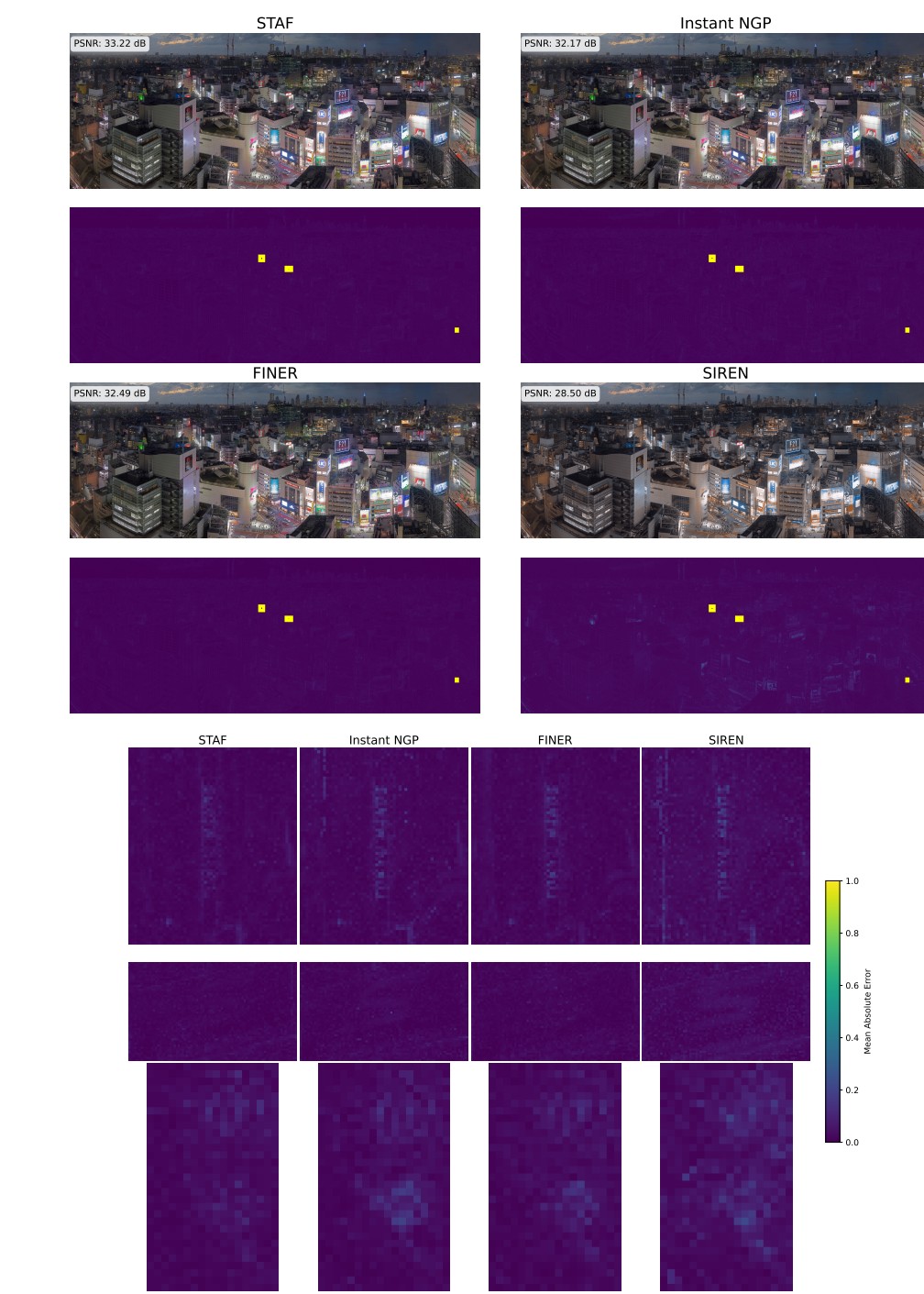

Figure 11: Comparative visualization of Tokyo image reconstruction and their error map using **STAF** and other activation functions. Yellow zoom-in regions are displayed in the last three rows.

## C.6 Performance Comparison of STAF and SIREN with Similar Parameter Counts

Figure 15 demonstrates the superior performance of STAF compared to SIREN in terms of PSNR (dB) across 250 epochs, despite SIREN having a higher parameter count for the Celtic image. To ensure a balanced evaluation, the default configuration of SIREN was modified by adding one additional layer, resulting in 264,193 parameters for SIREN compared to STAF's 213,761 parameters. This

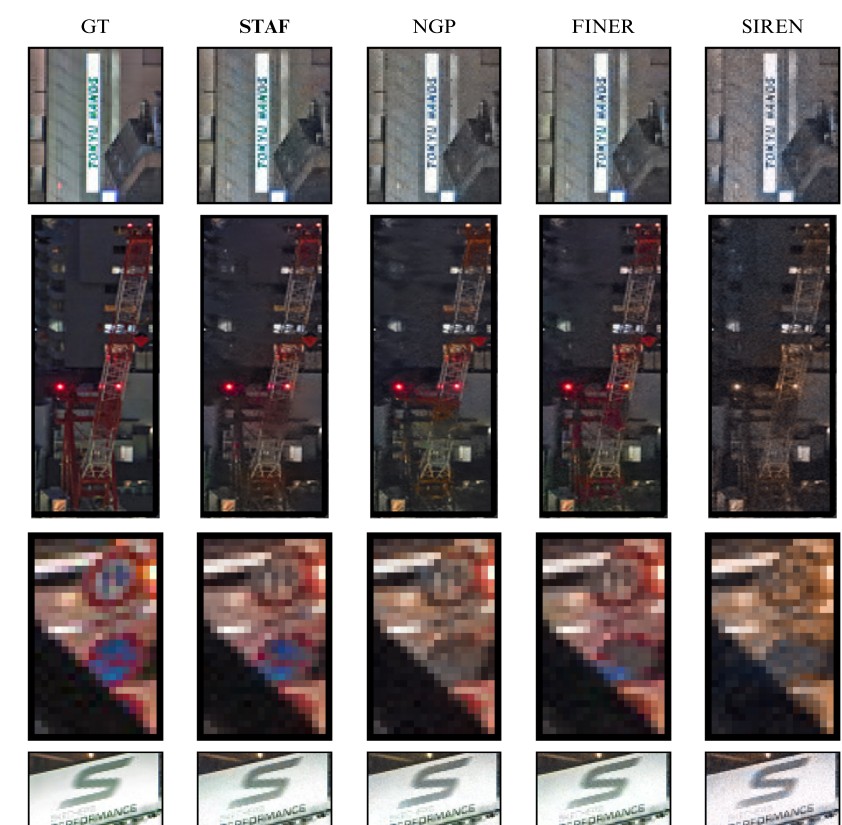

Figure 12: Reconstructed cropped patches of the Tokyo image using **STAF** and other activation functions.

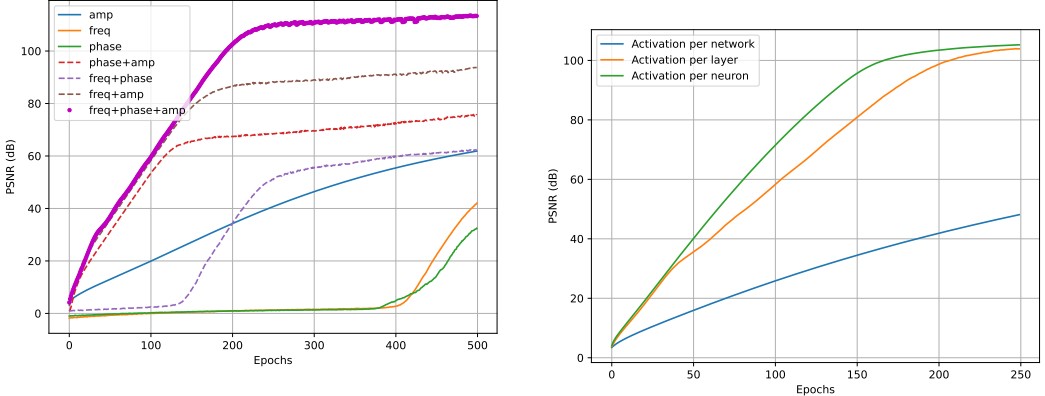

Figure 13: Ablation study of amplitude, frequency, and phase contributions on PSNR performance.

Figure 14: Analysis of activation patterns per network, layer, and neuron on PSNR performance.

approach avoids extensive parameter tuning for SIREN, offering a practical comparison between the two models. The results clearly show that STAF consistently outperforms SIREN, achieving significantly higher PSNR values throughout the training process. This highlights STAF's efficiency and effectiveness, even when constrained to a lower parameter count, making it a more suitable choice for tasks requiring high-quality image reconstruction.

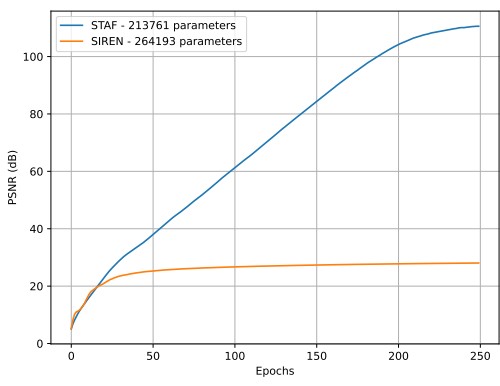

Figure 15: Comparison of PSNR performance between STAF and SIREN over 250 epochs. STAF, with 213,761 parameters, achieves significantly higher PSNR values compared to SIREN, which has 264,193 parameters.

Figure 16: Performance comparison of STAF, SIREN, and Hash Encoding on single image reconstruction. The PSNR curves show that STAF achieves the highest PSNR, followed by Hash Encoding and SIREN.

Table 5: Effect of varying $\tau$ on reconstruction quality, training, and inference time.

|  | $\tau = 2$ | $\tau = 5$ | $\tau = 10$ | $\tau = 20$ | $\tau = 50$ |
|---|---|---|---|---|---|
| **PSNR (dB) ↑** |  |  |  |  |  |
| Average | 37.07 | 38.80 | 39.14 | 41.27 | 41.41 |
| Variance | 20.99 | 13.58 | 13.44 | 12.74 | 7.63 |
| **SSIM ↑** |  |  |  |  |  |
| Average | 0.947 | 0.965 | 0.968 | 0.971 | 0.972 |
| Variance | 0.0022 | 0.0014 | 0.0011 | 0.0007 | 0.0001 |
| **LPIPS ↓** |  |  |  |  |  |
| Average | 0.021 | 0.012 | 0.010 | 0.006 | 0.005 |
| Variance | 0.0020 | 0.0002 | 0.0001 | 6.91e-5 | 5.18e-5 |
| **Training time (s)** | 47.38 | 92.89 | 162.43 | 325.53 | 742.94 |
| **Inference time (s)** | 0.079 | 0.157 | 0.273 | 0.556 | 1.268 |

Table 6: PSNR results on five randomly selected samples from the DIV2K 512×512 dataset.

| # | STAF | FINER | SIREN | FFN | PEMLP | WIRE | GAUSS |
|---|---|---|---|---|---|---|---|
| Sample 1 | **40.88** | 37.28 | 35.91 | 35.31 | 29.21 | 32.53 | 31.19 |
| Sample 2 | **34.93** | 34.90 | 30.81 | 31.56 | 23.42 | 31.28 | 30.47 |
| Sample 3 | **33.38** | 31.69 | 29.64 | 30.29 | 21.63 | 30.52 | 27.43 |
| Sample 4 | **40.23** | 38.01 | 36.94 | 35.19 | 25.32 | 32.89 | 31.65 |
| Sample 5 | **34.81** | 33.70 | 33.09 | 33.01 | 22.81 | 30.68 | 27.69 |

## C.7 MORE COMPARATIVE EVALUATIONS

Figure 16 presents a comparative analysis of three methods, STAF, SIREN, and Hash Encoding (Müller et al., 2022). The PSNR curves indicate that STAF significantly outperforms both SIREN and Hash Encoding, reaching a PSNR of over 100 dB after 500 epochs. While Hash Encoding shows a notable improvement over SIREN, peaking at around 70 dB, it still falls short of STAF's superior performance. SIREN, in contrast, exhibits the slowest PSNR growth, achieving only around 38 dB. The qualitative comparisons on the right further support these quantitative results, with STAF closely approximating the ground truth, while Hash Encoding and SIREN produce visibly lower-quality reconstructions. This analysis highlights the advantage of STAF in achieving both higher fidelity and faster convergence in image reconstruction tasks.

## C.8 MORE IMPLEMENTATION DETAILS

We provide the `PyTorch` implementation of our initialization scheme in Listing 1 to better illustrate our approach. We set $W_0$ (or `omega_0` in the code) to 30, and found that 40 also works well in practice. For the image representation task, we use a learning rate of $1e-3$. For shape representation, we use $2.5e-4$ with a marching cubes threshold of 0.5. In the audio task, we adopt a learning rate of $2.5e-4$, with $W_0 = 3000.0$ in the first layer and $W_0 = 30.0$ in the hidden layers. For denoising, we use a learning rate of $1.5e-4$ with $W_0 = 5$. Additionally, for high-resolution DIV2K images, we follow the approach in (Kazerouni et al., 2025) and incorporate skip connections in the hidden layers.

Table 7: Performance comparison across models in terms of parameter count, FLOPs, PSNR, SSIM, and LPIPS. FLOPs are calculated using the Calflops package (MrYxJ, 2023) with an input shape of (1, 1, 2).

| Model | #Parameters | #FW FLOPs | #Total (FW+BW) FLOPs | PSNR (dB) ↑ | SSIM ↑ | LPIPS ↓ |
|---|---|---|---|---|---|---|
| STAF ($\tau = 5$) | 198,975 | 395.78 K | 1.19 M | 39.94 | 0.973 | 0.005 |
| STAF ($\tau = 10$) | 199,035 | 395.78 K | 1.19 M | 40.84 | 0.979 | 0.003 |
| INCODE | 436,775 | 6.85 G | 20.55 G | 38.19 | 0.962 | 0.011 |
| FINER | 198,915 | 395.78 K | 1.19 M | 36.55 | 0.950 | 0.017 |
| WIRE | 99,915 | 198.38 K | 595.13 K | 32.76 | 0.885 | 0.058 |
| MFN | 204,291 | 398.85 K | 1.2 M | 33.87 | 0.942 | 0.020 |
| Gauss | 198,915 | 395.78 K | 1.19 M | 32.21 | 0.888 | 0.070 |
| FFN | 204,035 | 406.02 K | 1.22 M | 35.31 | 0.944 | 0.037 |
| SIREN | 198,915 | 395.78 K | 1.19 M | 33.55 | 0.928 | 0.062 |
| ReLU + PE | 206,083 | 411.14 K | 1.23 M | 30.10 | 0.856 | 0.150 |

Listing 1: Parameter initialization of STAF in PyTorch.

# D    PROOFS

## D.1    DISTRIBUTION OF $Y = \sin(aX + b)$ FOR $X \sim U(-1, 1)$

The authors of (Sitzmann et al., 2020) claim that regardless of the choice of $b$, if $a > \frac{\pi}{2}$, the output $Y$ follows an arcsine distribution, denoted as $Arcsine(-1, 1)$.

### D.1.1    ASSESSING SIREN'S CLAIM ABOUT THE DISTRIBUTION OF $Y = \sin(ax + b)$

If SIREN's claim were true, the expected value of $Y$ would be independent of $b$.

$$\mathbb{E}[Y] = \int_{-1}^{1} \sin(ax + b) f_X(x) \, dx = \frac{1}{2} \int_{-1}^{1} (\sin(ax) \cos b + \sin b \cos(ax)) \, dx$$

$$= \frac{1}{2a} \Big[ \big( \cos(-a) - \cos(a) \big) \cos b + \big( \sin(a) - \sin(-a) \big) \sin b \Big] = \frac{\sin a \sin b}{a} \tag{36}$$

which obviously depends on $a$ and $b$. However, if we want to eliminate $b$ from $\mathbb{E}[Y]$, we can set

$$a = n\pi, \tag{37}$$

for an $n \in \mathbb{Z} \setminus \{0\}$. Next, let us consider the next moments of $Y$, because if the moment-generating function (MGF) of $Y$ exists, the moments can uniquely determine the distribution of $Y$.

$$\mathbb{E}[Y^k] = \int_{-1}^{1} \frac{1}{2} \sin^k(ax + b) dx \tag{38}$$

Using equation 37, it is equal to $\frac{1}{2n\pi} \int_{-1}^{1} \sin^k(ax + b) dx$. By assuming $u = ax + b$, we have

$$\mathbb{E}[Y^k] = \frac{1}{2n\pi} \int_{b-a}^{b-a+2n\pi} \sin^k(u) du. \tag{39}$$

Since for each pair of natural numbers $(k, n)$, $2n\pi$ is a period of $\sin^k(u)$, we can write

$$\mathbb{E}[Y^k] = \frac{1}{2n\pi} \int_{0}^{2\pi} \sin^k(u) du = \begin{cases} 0, & \text{if } k \text{ is odd} \\ \frac{\binom{k}{k/2}}{2^k n}, & \text{if } k \text{ is even} \end{cases} \tag{40}$$

If $a = n\pi$, then all moments of $Y$ will be independent of $a$ and $b$. Consequently, one may expect that the distribution of $Y$ is also independent of $a$ and $b$. However, this only occurs when $a = n\pi$. Therefore, the assumption that the distribution of $Y$ is independent of the parameter $b$ (as used in (Sitzmann et al., 2020)) holds only for specific values of $a$.

We next calculate the exact distribution of $Y = \sin(aX + b)$, where $X \sim U(-1, 1)$.

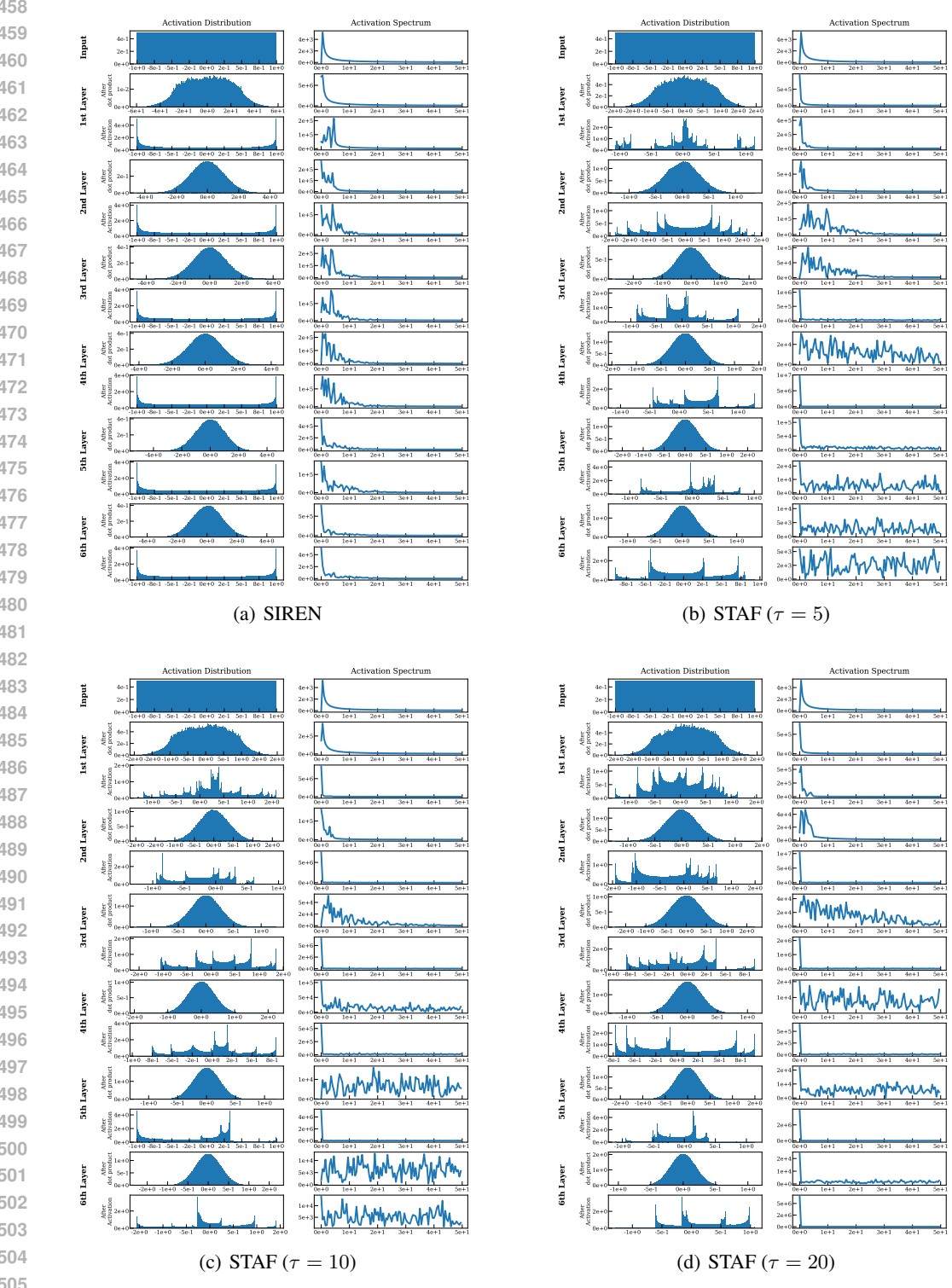

Figure 17: Comparison of activation distributions and frequency spectra between SIREN and STAF with $\tau$ values of 5, 10, and 20. Note that the scales may differ across plots; however, to appreciate how effectively STAF captures higher frequencies, observe the maximum frequency in each spectrum.

### D.1.2 FINDING THE DISTRIBUTION OF $Y$

To obtain the distribution of the random variable

$$Y = \sin(aX + b)$$

we use the change-of-variables (Jacobian) method. First note that the probability density function of $X$ is

$$f_X(x) = \begin{cases} \dfrac{1}{2}, & -1 \le x \le 1, \\ 0, & \text{otherwise.} \end{cases} \tag{41}$$

Consider the transformation

$$Y = g(X) = \sin(aX + b). \tag{42}$$

To obtain the density of $Y$ we need to find all possible inverses of $g$. If $y = \sin(ax + b)$ then

$$ax + b = \arcsin(y) + 2k\pi \quad \text{or} \quad ax + b = \pi - \arcsin(y) + 2k\pi, \qquad k \in \mathbb{Z}. \tag{43}$$

Hence

$$x = \frac{\arcsin(y) - b + 2k\pi}{a} \quad \text{or} \quad x = \frac{\pi - \arcsin(y) - b + 2k\pi}{a}. \tag{44}$$

The density of $Y$ is given by

$$f_Y(y) = \sum_{x \in g^{-1}(y)} f_X(x) \left| \frac{dx}{dy} \right|. \tag{45}$$

Since

$$\frac{dy}{dx} = a\cos(ax + b), \tag{46}$$

we have

$$\frac{dx}{dy} = \frac{1}{a\cos(ax + b)}, \tag{47}$$

and therefore

$$f_Y(y) = \frac{1}{2} \sum_{x \in g^{-1}(y) \cap [-1,1]} \frac{1}{|a\cos(ax + b)|}. \tag{48}$$

Because at any root $x$ of $\sin(ax + b) = y$ we have $\cos(ax + b) = \pm\sqrt{1 - y^2}$, each summand is independent of the particular root and depends only on $y$. Thus

$$f_Y(y) = \frac{1}{2} \sum_{x \in g^{-1}(y) \cap [-1,1]} \frac{1}{|a|\sqrt{1 - y^2}}. \tag{49}$$

Let $N(y)$ denote the number of solutions $x \in [-1, 1]$ of the equation $\sin(ax + b) = y$. Then

$$\boxed{f_Y(y) = \begin{cases} \dfrac{N(y)}{2|a|\sqrt{1 - y^2}}, & y \in \mathcal{I} := \{\sin t : t \in [b - a, b + a]\}, \\ 0, & \text{otherwise,} \end{cases}}$$

where $\mathcal{I}$ is the image of the interval $[-1, 1]$ under the map $t = ax + b \mapsto \sin t$ (note $\mathcal{I} \subseteq [-1, 1]$).

One can also write $N(y)$ explicitly as the count of integer indices $k$ giving roots in $[-1, 1]$:

$$N(y) = \#\left\{ k \in \mathbb{Z} : \frac{\arcsin y + 2k\pi - b}{a} \in [-1, 1] \right\} + \#\left\{ k \in \mathbb{Z} : \frac{\pi - \arcsin y + 2k\pi - b}{a} \in [-1, 1] \right\}. \tag{50}$$

In practice, to determine $N(y)$ one typically considers the interval $[b - a, b + a]$ and, for each $y$, counts the number of points $t$ in that interval such that $\sin t = y$. For example:

- If the length of the interval $b + a - (b - a) = 2|a|$ is less than $\pi$ and the interval is positioned so that $\cos t$ does not change sign there, then $\sin t$ is monotone on that interval and hence $N(y) = 1$. In this case the density simplifies to

$$f_Y(y) = \frac{1}{2|a|\sqrt{1 - y^2}}, \qquad y \in \Big[ \min_{t \in [b-a, b+a]} \sin t, \ \max_{t \in [b-a, b+a]} \sin t \Big]. \qquad (51)$$

- If the interval $[b - a, b + a]$ contains more than one oscillation of the sine (for instance if its length exceeds $2\pi$, or it contains several peaks and troughs), then $N(y)$ can be greater than 1. Inside the interior of the support, $N(y)$ is typically constant and changes in integer steps (blockwise).

Let $m$ and $M$ denote the minimum and maximum of $\sin t$ on $[b - a, b + a]$, respectively. In the special case $\frac{\pi}{2} \leq a < \pi$, by the first case above we obtain

$$f_Y(y) = \frac{1}{2a\sqrt{1 - y^2}}, \qquad y \in [m, M]. \qquad (52)$$

Consequently, the cumulative distribution function of $Y$ is

$$F_Y(y) = \int_{-\infty}^{y} \frac{dt}{2a\sqrt{1 - t^2}} = \begin{cases} 0, & y < m, \\ \frac{1}{2a}\big(\arcsin(y) - \arcsin(m)\big), & m \leq y \leq M, \\ 1, & y > M. \end{cases} \qquad (53)$$

As is evident, since the distribution of $Y$ depends on $m$ and $M$, it depends on $b$ even in this case. Therefore, the assumption in Sitzmann et al. (2020) that when $a > \frac{\pi}{2}$ the distribution of $Y$ is independent of $b$ is not correct.

### D.2 Proof of Theorem equation 3.1

In this section, we provide a step-by-step proof of Theorem equation 3.1 concerning the initialization scheme of an architecture that leverages STAF.

**Theorem D.1.** *Consider the function $Z$ defined as*

$$Z = \sum_{u=1}^{\tau} C_u \sin\left(\Gamma_u + \Phi_u\right). \qquad (54)$$

*Suppose that the random variables $C_u$ have symmetric distributions and finite moments. Moreover, for each $u$, assume that $\Phi_u \sim U(-\pi, \pi)$. In addition, assume the following independence conditions:*

- *the variables $C_i$ are mutually independent;*

- *for each $i$, the variable $C_i$ is independent of $(\Gamma_i, \Phi_i)$; and*

- *the collections $\{(C_i, \Gamma_i, \Phi_i)\}_{i=1}^{\tau}$ are mutually independent.*

*Then, the moments of $Z$ depend only on $\tau$ and the moments of the $C_u$'s, and all odd-order moments of $Z$ will be zero.*

*Proof.* For convenience, let us consider $\Gamma_u = \Omega_u \boldsymbol{w}.\boldsymbol{x}$. Based on the multinomial theorem, for every natural number $q$, we have:

$$Z^q = \sum_{\substack{i_1 + \ldots + i_\tau = q \\ i_1, \ldots, i_\tau \geq 0}} \left[ \binom{q}{i_1, \ldots, i_\tau} \prod_{u=1}^{\tau} \left(C_u \sin(\Gamma_u + \Phi_u)\right)^{i_u} \right].$$

According to the linearity of expected value:

$$\mathbb{E}[Z^q] = \sum_{\substack{i_1+\ldots+i_\tau=q \\ i_1,\ldots,i_\tau \geq 0}} \left[ \binom{q}{i_1,\ldots,i_\tau} \mathbb{E}\left[ \prod_{u=1}^{\tau} \left( C_u \sin(\Gamma_u + \Phi_u) \right)^{i_u} \right] \right] \tag{55}$$

Given the independence assumptions stated above:

$$\mathbb{E}[Z^q] = \sum_{\substack{i_1+\ldots+i_\tau=q \\ i_1,\ldots,i_\tau \geq 0}} \left[ \binom{q}{i_1,\ldots,i_\tau} \prod_{u=1}^{\tau} \left[ \mathbb{E}[C_u^{i_u}] \mathbb{E}\left[ \sin^{i_u}(\Gamma_u + \Phi_u) \right] \right] \right]. \tag{56}$$

Each choice of $i_1, \ldots, i_\tau$ is called a partition for $q$. If $q$ is an odd number, then in each partition of $q$, at least one of the variables, such as $i_k$, is odd. Since the function $C_i$ is symmetric, it follows that $\mathbb{E}[C_k^{i_k}] = 0$. This is because odd-order moments of a symmetric distribution are always zero. Consequently, the expectation $\mathbb{E}\left[ \prod_{u=1}^{\tau} \left( C_u \sin(\Gamma_u + \Phi_u) \right)^{i_u} \right]$ also equals zero, as does $\mathbb{E}[Z^q]$.

Now, let us consider the case when $q$ is even. For each partition of $q$, if at least one of its variables is odd, then, as before, we have $\mathbb{E}\left[ \prod_{u=1}^{\tau} \left( C_u \sin(\Gamma_u + \Phi_u) \right)^{i_u} \right] = 0$. Thus, we can express $q$ as $q = 2j_1 + \ldots + 2j_\tau$ where each $j_k$ is a non-negative integer. According to equation 56, to obtain the $i_k$-th moment of $Z$, we need to calculate $\mathbb{E}\left[ \sin^{i_u}(\Gamma_u + \Phi_u) \right]$. In this case, where $i_u = 2j_u$, $\sin^{i_u} \theta$ is an even function, and its Fourier series consists of a constant term and some cosine terms, given by

$$\sin^{2j_u} \theta = \alpha_0 + \sum_{r=1}^{\infty} \alpha_r \cos(r\theta). \tag{57}$$

Hence,

$$\mathbb{E}[\sin^{2j_u}(\Gamma_u + \Phi_u)] = \mathbb{E}[\alpha_0 + \sum_{r=1}^{\infty} \alpha_r \cos(r(\Gamma_u + \Phi_u))] = \alpha_0 + \sum_{r=1}^{\infty} \alpha_r \mathbb{E}[\cos(r\Gamma_u + r\Phi_u)]$$

$$= \alpha_0 + \sum_{r=1}^{\infty} \alpha_r \mathbb{E}[\cos(r\Gamma_u)\cos(r\Phi_u) - \sin(r\Gamma_u)\sin(r\Phi_u)] = \alpha_0 + \sum_{r=1}^{\infty} \alpha_r \Xi \tag{58}$$

where

$$\Xi = \mathbb{E}[\cos(r\Gamma_u)]\mathbb{E}[\cos(r\Phi_u)] - \mathbb{E}[\sin(r\Gamma_u)]\mathbb{E}[\sin(r\Phi_u)]. \tag{59}$$

Since $r$ is an integer, $r\Phi_u$ will be a period, resulting in $\mathbb{E}[\cos(r\Phi_u)] = \mathbb{E}[\sin(r\Phi_u)] = 0$. Thus, $\mathbb{E}[\sin^{2j_u}(\Gamma_u + \Phi_u)] = \alpha_0$.

Using the formula for the coefficients of the Fourier series, we have:

$$\alpha_0 = \frac{1}{\pi} \int_{-\pi/2}^{\pi/2} \sin^{2j_u} \theta \, d\theta = \frac{2}{\pi} \int_0^{\pi/2} \sin^{2j_u} \theta \, d\theta = \frac{2}{\pi} \times \frac{\binom{2j_u}{j_u}}{2^{2j_u}} \times \frac{\pi}{2} = \frac{\binom{2j_u}{j_u}}{2^{2j_u}} \tag{60}$$

where equation 60 is evaluated using the Wallis integral.

To summarize,

$$\mathbb{E}[Z^q] = \sum_{\substack{j_1+\cdots+j_\tau=\frac{q}{2}, \\ j_1,\ldots,j_\tau \geq 0}} \binom{q}{2j_1,\ldots,2j_\tau} \prod_{u=1}^{\tau} \mathbb{E}[C_u^{2j_u}] \frac{\binom{2j_u}{j_u}}{2^{2j_u}}$$

$$= \sum_{\substack{j_1+\cdots+j_\tau=\frac{q}{2}, \\ j_1,\ldots,j_\tau \geq 0}} \left[ \left( \binom{q}{2j_1,\ldots,2j_\tau} \prod_{u=1}^{\tau} \binom{2j_u}{j_u} \right) \prod_{u=1}^{\tau} \frac{1}{2^{2j_u}} \prod_{u=1}^{\tau} \mathbb{E}[C_u^{2j_u}] \right] \tag{61}$$

This also accounts for odd-order moments, as it is impossible to select a combination of non-negative integers that sums to a non-integer value.

It is worth noting that:

$$\binom{q}{2j_1, \ldots, 2j_\tau} \prod_{u=1}^{\tau} \binom{2j_u}{j_u} = \frac{q!}{(2j_1)! \ldots (2j_\tau)!} \times \frac{(2j_1)!}{(j_1)!^2} \times \cdots \times \frac{(2j_\tau)!}{(j_\tau)!^2} = \frac{q!}{(j_1!)^2 \ldots (j_\tau!)^2}$$
$$= \binom{q}{j_1, j_1, \ldots, j_\tau, j_\tau} \tag{62}$$

Furthermore,

$$\prod_{u=1}^{\tau} \frac{1}{2^{2j_u}} = \frac{1}{2^{2 \sum_{u=1}^{\tau} j_u}} = \frac{1}{2^q} \tag{63}$$

By utilizing Equations equation 61 to equation 63, we can conclude that:

$$\mathbb{E}[Z^q] = \frac{1}{2^q} \sum_{\substack{j_1 + \cdots + j_\tau = \frac{q}{2} \\ j_1, \ldots, j_\tau \geq 0}} \binom{q}{j_1, j_1, \ldots, j_\tau, j_\tau} \prod_{u=1}^{\tau} \mathbb{E}[C_u^{2j_u}] \tag{64}$$

As you can see, the moments of $Z$ depend solely on $\tau$ and the moments of the $C_u$'s. $\square$

Now, our goal is to determine the distribution of the $C_u$'s so that the distribution of $Z$ becomes $\mathcal{N}(0, 1)$. To achieve this, let's first consider the following theorem:

**Theorem D.2.** *(Page 353 of (Shiryaev, 2016)) Let $X \sim \mathcal{N}(0, \sigma^2)$. Then*

$$E(X^q) = \begin{cases} 0, & \text{if } q \text{ is odd} \\ \frac{q!}{\frac{q}{2}! \, 2^{q/2}} \sigma^q, & \text{if } q \text{ is even} \end{cases} \tag{65}$$

*and these moments pertain exclusively to the normal distribution.*

In theorem equation D.1, we proved that for odd values of $q$, $\mathbb{E}[h^q] = 0$. Thus, in order to have $Z \sim \mathcal{N}(0, 1)$, for even values of $q$, we must have $\mathbb{E}[h^q] = \frac{q!}{\frac{q}{2}! \, 2^{q/2}}$. Alternatively, we can express it as

$$\frac{1}{2^q} \sum_{\substack{j_1 + \cdots + j_\tau = \frac{q}{2} \\ j_1, \ldots, j_\tau \geq 0}} \binom{q}{j_1, j_1, \ldots, j_\tau, j_\tau} \prod_{u=1}^{\tau} \mathbb{E}[C_u^{2j_u}] = \frac{q!}{\frac{q}{2}! \, 2^{q/2}}. \tag{66}$$

Simplifying further, we obtain

$$\frac{q!}{2^q} \sum_{\substack{j_1 + \cdots + j_\tau = \frac{q}{2} \\ j_1, \ldots, j_\tau \geq 0}} \frac{\prod_{u=1}^{\tau} \mathbb{E}[C_u^{2j_u}]}{(j_1!)^2 \ldots (j_\tau!)^2} = \frac{q!}{\frac{q}{2}! \, 2^{q/2}}. \tag{67}$$

This equation can be further simplified to

$$\sum_{\substack{j_1 + \cdots + j_\tau = \frac{q}{2} \\ j_1, \ldots, j_\tau \geq 0}} \frac{\prod_{u=1}^{\tau} \mathbb{E}[C_u^{2j_u}]}{(j_1!)^2 \ldots (j_\tau!)^2} = \frac{2^{q/2}}{\frac{q}{2}!}. \tag{68}$$

Equation equation 68 provides a general formula that can be utilized in further research. It allows for finding different solutions for $C_u$ under various assumptions (e.g., independence or specific dependencies) and different values of $\tau$. However, in the subsequent analysis, we assume that $C_u$'s are independent and identically distributed (i.i.d) random variables. The following theorem aims to satisfy Equation equation 68.

**Theorem D.3.** *Suppose $C_u$'s are i.i.d random variables with the following even-order moments:*

$$\mathbb{E}[C_u^{2j}] = \left(\frac{2}{\tau}\right)^j j! \tag{69}$$

*Then, for every non-negative even number $q$, Equation equation 68 holds.*[3]

---

[3]If you wonder how this solution struck our mind, you can start by solving equation equation 68 for $q = 2$ to obtain $\mathbb{E}[h^2]$. Then, using the value of $\mathbb{E}[h^2]$, solve equation 68 for $q = 4$ to obtain $\mathbb{E}[h^4]$, and so on.

*Proof.* We begin by simplifying the expression:

$$
\sum_{\substack{j_1+\cdots+j_\tau=\frac{q}{2} \\ j_1,\ldots,j_\tau \geq 0}} \frac{\prod_{u=1}^{\tau} \mathbb{E}[C_u^{2j_u}]}{(j_1!)^2\ldots(j_\tau!)^2} = \sum_{\substack{j_1+\cdots+j_\tau=\frac{q}{2} \\ j_1,\ldots,j_\tau \geq 0}} \frac{\prod_{u=1}^{\tau}\left[\left(\frac{2}{\tau}\right)^j j!\right]}{(j_1!)^2\ldots(j_\tau!)^2}
$$

$$
= \sum_{\substack{j_1+\cdots+j_\tau=\frac{q}{2} \\ j_1,\ldots,j_\tau \geq 0}} \left(\frac{2}{\tau}\right)^{\sum_{u=1}^{\tau} j_u} \left(\frac{1}{j_1!\ldots j_\tau!}\right) = \sum_{\substack{j_1+\cdots+j_\tau=\frac{q}{2} \\ j_1,\ldots,j_\tau \geq 0}} \left(\frac{2}{\tau}\right)^{\frac{q}{2}} \left(\frac{1}{j_1!\ldots j_\tau!}\right)
$$

$$
= \left(\frac{2}{\tau}\right)^{\frac{q}{2}} \sum_{\substack{j_1+\cdots+j_\tau=\frac{q}{2} \\ j_1,\ldots,j_\tau \geq 0}} \frac{1}{j_1!\ldots j_\tau!} = \left(\frac{2}{\tau}\right)^{\frac{q}{2}} \frac{1}{(\frac{q}{2})!} \sum_{\substack{j_1+\cdots+j_\tau=\frac{q}{2} \\ j_1,\ldots,j_\tau \geq 0}} \frac{(\frac{q}{2})!}{j_1!\ldots j_\tau!}
$$

$$
= \left(\frac{2}{\tau}\right)^{\frac{q}{2}} \frac{1}{(\frac{q}{2})!} \sum_{\substack{j_1+\cdots+j_\tau=\frac{q}{2} \\ j_1,\ldots,j_\tau \geq 0}} \binom{\frac{q}{2}}{j_1,\ldots,j_\tau} \tag{70}
$$

Based on the multinomial theorem, we can conclude that

$$
\left(\frac{2}{\tau}\right)^{\frac{q}{2}} \frac{1}{(\frac{q}{2})!} \sum_{\substack{j_1+\cdots+j_\tau=\frac{q}{2} \\ j_1,\ldots,j_\tau \geq 0}} \binom{\frac{q}{2}}{j_1,\ldots,j_\tau} = \left(\frac{2}{\tau}\right)^{\frac{q}{2}} \frac{\tau^{\frac{q}{2}}}{(\frac{q}{2})!} = \frac{2^{\frac{q}{2}}}{(\frac{q}{2})!} \tag{71}
$$

$\square$

Also note that according to Theorem equation D.1, the odd-order moments of $Z$ are zero, just like a normal distribution.

**Corollary D.4.** *Let $Z$ be the random variable defined in equation 54. Additionally, assume that the $C_u$'s ($1 \leq u \leq \tau$) used in the definition of $Z$, are i.i.d random variables with even moments as defined in theorem equation D.3. Then $Z \sim \mathcal{N}(0,1)$.*

*Proof.* We know that if the MGF of a distribution exists, then the moments of that distribution can uniquely determine its PDF. That is, if $X$ and $Y$ are two distributions and for every natural number $k$, $E(X^k) = E(Y^k)$, then $X = Y$.

In the Theorem equation D.3, we observed that the moments of $Z$ are equal to the moments of a standard normal distribution. Since the MGF of this distribution exists, $Z \sim \mathcal{N}(0,1)$. $\square$

Now, let's explore which distribution can produce the moments defined in equation equation 69. To have an inspiration, note that for a centered Laplace random variable $X$ with scale parameter $b$, we have the PDF of $X$ as

$$
f_X(x) = \frac{1}{2b} e^{\frac{-|x|}{b}} \tag{72}
$$

and the moments of $X$ given by

$$
\mathbb{E}[X^q] = \begin{cases} 0, & \text{if } q \text{ is odd} \\ \frac{b^q}{q!}, & \text{if } q \text{ is even} \end{cases} \tag{73}
$$

Hence, the answer might be similar to this distribution. If we assume $Y = sgn(X)\sqrt{|X|}$, since $Y$ is symmetric, all of its odd-order moments are zero. Now, let us calculate its even-order moments:

$$
\mathbb{E}[Y^{2q}] = \mathbb{E}[|X|^q] = \int_{-\infty}^{\infty} |x|^q \frac{1}{2b} e^{-\frac{|x|}{b}} dx = 2\int_0^{\infty} |x|^q \frac{1}{2b} e^{-\frac{|x|}{b}} dx = \frac{1}{b}\int_0^{\infty} x^q e^{-\frac{x}{b}} dx \tag{74}
$$

By assuming $u = \frac{x}{b}$, we will have

$$
\mathbb{E}[Y^{2q}] = \int_0^{\infty} (bu)^q e^{-u} du = b^q \int_0^{\infty} u^q e^{-u} du = b^q \Gamma(q+1) = b^q q! \tag{75}
$$

By assuming $b = \frac{2}{\tau}$, equation 69 will be obtained.

The next theorem will obtain the probability distribution function of $Y$.

**Theorem D.5.** *Let $X$ be a centered Laplace random variable with scale parameter $b$, and $Y = sgn(X)\sqrt{|X|}$. Then*

$$f_Y(y) = \frac{|y|}{b} e^{\frac{-y^2}{b}} \tag{76}$$

*Proof.* Let $A = Y^2 = |X|$. Therefore,

$$M_A(t) = \sum_{k=0}^{\infty} \frac{t^k \mathbb{E}[|X|^k]}{k!} \tag{77}$$

As we calculated in equation 75, $\mathbb{E}[|X|^k] = b^k \, k!$. Therefore,

$$M_A(t) = \sum_{k=0}^{\infty} \frac{t^k \cdot b^k \, k}{k!} = \sum_{k=0}^{\infty} (bt)^k = \frac{1}{1 - bt} = \frac{\frac{1}{b}}{\frac{1}{b} - t} \tag{78}$$

that is the MGF of exponential distribution with parameter $\frac{1}{b}$. That is,

$$f_A(a) = \frac{1}{b} e^{\frac{-a}{b}} \tag{79}$$

Therefore, using the fact that $A$ is a always non-negative, we consider non-negative values $a^2$ to describe its cumulative distribution function.

$$F_A(y^2) = \mathbb{P}(A \le y^2) = 1 - e^{\frac{-y^2}{b}} \tag{80}$$

On the other hand, if $y \ge 0$,

$$\mathbb{P}(A \le y^2) = \mathbb{P}(Y^2 \le y^2) = \mathbb{P}(-y \le Y \le y) \tag{81}$$

Since we want $Y$ to be symmetric, we assume[4]

$$\mathbb{P}(-y \le Y \le y) = 2\,\mathbb{P}(0 \le Y \le y) = 2\left(\mathbb{P}(Y \le y) - \frac{1}{2}\right) = 2F_Y(y) - 1, \quad y \ge 0 \tag{82}$$

Using equations equation 80 to equation 82, we draw conclusion that

$$2F_Y(y) - 1 = 1 - e^{\frac{-y^2}{b}}, \quad y \ge 0 \tag{83}$$

By differentiating both sides of equation 83 with respect to $y$, we will have

$$2f_Y(y) = \frac{2y}{b} e^{\frac{-y^2}{b}}, \quad y \ge 0 \tag{84}$$

Therefore,

$$f_Y(y) = \frac{y}{b} e^{\frac{-y^2}{b}}, \quad y \ge 0 \tag{85}$$

Since we assumed $y \ge 0$ in the above equations, and we supposed that $Y$ is symmetric,

$$f_Y(y) = \frac{|y|}{b} e^{\frac{-y^2}{b}}, \quad y \in \mathbb{R} \tag{86}$$

Just to make sure that our assumption about the symmetry of $Y$ was correct (or sufficed for our purpose), let us check the even-order moments of $Y$. The odd-orders are zero based on the symmetry.

$$\mathbb{E}[Y^{2k}] = \int_{-\infty}^{\infty} y^{2k} \left(\frac{|y|}{b} e^{-\frac{y^2}{b}}\right) dy = \frac{2}{b} \int_0^{\infty} y^{2k+1} e^{-\frac{y^2}{b}} dy \tag{87}$$

Setting $y^2 = t$ and $\frac{1}{b} = s$, leads to the following equation:

$$\mathbb{E}[Y^{2k}] = \frac{1}{b} \int_0^{\infty} t^k e^{-st} dt \tag{88}$$

That is the Laplace transform of $t^k$. Therefore,

$$\mathbb{E}[Y^{2k}] = s\frac{\Gamma(k+1)}{s^{k+1}} = \frac{k!}{s^k} = b^k k! \tag{89}$$

□

---

[4]In fact, the assumption that $Y$ is symmetric is not unexpected, since all odd-order moments of $Y$ are zero. But there are some non-symmetrical distributions whose all odd-order moments are zero (Churchill, 1946). Nevertheless, under some assumptions, it can be shown that a distribution is symmetric if and only if all its odd-order moments are zero. However, we don't use this claim in this paper.

In summary, in this section we calculated the initial coefficients of our activation function as described in Theorem equation D.5, where we set $b = \frac{2}{\tau}$. Consequently, if we denote the post-activation of layer $l$ by $z^{(l)}$, we will have $z_i^{(l)} \sim \mathcal{N}(0,1)$ for all $l \in \{2, 3, \ldots, L-1\}$, and $i \in \{1, \ldots F_l\}$. This result can be proved by induction on $l$, using the fact that, based on the theorems in this section, the PDF of $Z$ is independent of the PDF of $x$.

D.3 PROOF OF THEOREM EQUATION 4.2

Before proving the theorem, note the following remark:

*Remark* D.6. Let $X$ be a $\chi_1 \times \chi_2$ matrix, and $Y$ be a $\gamma_1 \times \gamma_2$ matrix. Then, according to (Ashendorf et al., 2014; Albrecht et al., 2023):

$$(X \otimes Y)_{i,j} = x_{\lceil i/\gamma_1 \rceil, \lceil i/\gamma_2 \rceil} \ y_{(i-1)\%\gamma_1+1, \ (j-1)\%\gamma_2+1}. \tag{90}$$

Now, let us consider each pair of layers as a block, where the first two layers form the first block, the second two layers form the second block, and so on. We prove the theorem by induction on the block numbers. The proof consists of three parts:

**Part 1)** Consider the weight matrix and bias vector given by:

$$\overline{W^{(l)}} = \Omega \otimes W^{(l)}, \qquad \overline{B^{(l)}} = \Phi \otimes J_{F_l,1}. \tag{91}$$

We then define

$$\begin{bmatrix} \overline{a_1^{(l)}} & \overline{a_2^{(l)}} & \ldots & \overline{a_{\tau F_l}^{(l)}} \end{bmatrix}^{tr} = \overline{W^{(l)}} z^{(l-1)} + \overline{B^{(l)}}, \tag{92}$$

and

$$\overline{z_p^{(l)}} = \rho(\overline{a_p^{(l)}}) \quad \forall \, p \in \{1, 2, \ldots, \tau F_l\}. \tag{93}$$

Additionally, define

$$\tilde{a}^{(l+1)} = \left( C^{tr} \otimes W_{i,:}^{(l+1)} \right) \overline{z^{(l)}}, \tag{94}$$

where $W_{i,:}^{(l+1)}$ denotes the $i$'th row of $W^{(l+1)}$. Then, we can observe that

$$\tilde{a}^{(l+1)} = a_i^{(l+1)} \tag{95}$$

*Proof.* First, let us calculate $a_i^{(l+1)}$ using activation function $\rho^*$. Note that $a^{(l+1)} = W^{(l+1)} z^{(l)}$. Therefore, $a_i^{(l+1)} = W_{i,:}^{(l+1)} z^{(l)}$. It implies that

$$a_i^{(l+1)} = \sum_{j=1}^{F_l} W_{i,j}^{(l+1)} z_j^{(l)} = \sum_{j=1}^{F_l} W_{i,j}^{(l+1)} \rho^* \left( a_j^{(l)} \right) = \sum_{j=1}^{F_l} W_{i,j}^{(l+1)} \rho^* \left( \sum_{k=1}^{F_{l-1}} W_{j,k}^{(l)} z_k^{(l-1)} \right)$$

$$= \sum_{j=1}^{F_l} W_{i,j}^{(l+1)} \sum_{m=1}^{\tau} C_m \rho \left( \Omega_m \sum_{k=1}^{F_{l-1}} W_{j,k}^{(l)} z_k^{(l-1)} + \Phi_m \right) \tag{96}$$

Next, let us calculate $\tilde{a}^{(l+1)}$. We have

$$\overline{a_p^{(l)}} = \left[ \overline{W^{(l)}} z^{(l-1)} + \overline{B^{(l)}} \right]_p = \overline{W^{(l)}}_{p,:} z^{(l-1)} + \overline{B^{(l)}}_p = \sum_{k=1}^{F_{l-1}} \left( \overline{W^{(l)}}_{p,k} z_k^{(l-1)} \right) + \overline{B^{(l)}}_p$$

$$= \sum_{k=1}^{F_{l-1}} \left( \Omega_{\lceil p/F_l \rceil, \lceil k/F_{l-1} \rceil} W_{1+(p-1)\%F_l, 1+(k-1)\%F_{l-1}}^{(l)} z_k^{(l-1)} \right) + \Phi_{\lceil p/F_l \rceil} \tag{97}$$

Equation equation 97 is based on equation equation 90. Since $1 \le k \le F_{l-1}$, it follows that $\lceil k/F_{l-1} \rceil = 1$ and $(k-1)\%F_{l-1} = k - 1$. As a result,

$$\overline{a_p^{(l)}} = \sum_{k=1}^{F_{l-1}} \left( \Omega_{\lceil p/F_l \rceil} W_{1+(p-1)\%F_l, k}^{(l)} z_k^{(l-1)} \right) + \Phi_{\lceil p/F_l \rceil}$$

$$= \Omega_{\lceil p/F_l \rceil} \sum_{k=1}^{F_{l-1}} \left( W_{1+(p-1)\%F_l, k}^{(l)} z_k^{(l-1)} \right) + \Phi_{\lceil p/F_l \rceil} \tag{98}$$

Therefore,

$$\overline{z_p^{(l)}} = \rho \left( \boldsymbol{\Omega}_{\lceil p/F_l \rceil} \sum_{k=1}^{F_{l-1}} \left( W_{1+(p-1)\%F_l,k}^{(l)} z_k^{(l-1)} \right) + \boldsymbol{\Phi}_{\lceil p/F_l \rceil} \right) \tag{99}$$

Consequently,

$$\tilde{a}^{(l+1)} = \sum_{p=1}^{\tau F_l} \left[ \boldsymbol{C}^{tr} \otimes W_{i,:}^{(l+1)} \right]_{1,p} \overline{z_p^{(l)}} = \sum_{p=1}^{\tau F_l} \boldsymbol{C}_{1,\lceil p/F_l \rceil}^{tr} W_{i,1+(p-1)\%F_l}^{(l+1)} \overline{z_p^{(l)}}$$

$$= \sum_{p=1}^{\tau F_l} \boldsymbol{C}_{\lceil p/F_l \rceil} W_{i,1+(p-1)\%F_l}^{(l+1)} \overline{z_p^{(l)}} \tag{100}$$

$$= \sum_{j=1}^{F_l} \sum_{m=1}^{\tau} \boldsymbol{W}_{i,j}^{(l+1)} \boldsymbol{C}_m \overline{z_{F_l(m-1)+j}^{(1)}} \tag{101}$$

Equation equation 101 is obtained as follows: by changing the indices of $\boldsymbol{W}$ and $\boldsymbol{C}$ from equation equation 100 to equation 101, we need to change the index of $z^{(l)}$ too. To this end, note that

$$m = \lceil p/F_l \rceil, \quad j = 1 + (p-1)\%F_l \tag{102}$$

If $F_l \nmid p$, then $m = 1 + \lfloor p/F_l \rfloor$. As we know, $p = F_l \lfloor p/F_l \rfloor + p\%F_l$. Therefore, $p = F_l(m-1)+j$. This equation also holds when $F_l \mid p$.

Equation equation 101 can be rewritten as follows:

$$\sum_{j=1}^{F_l} \boldsymbol{W}_{i,j}^{(l+1)} \sum_{m=1}^{\tau} \boldsymbol{C}_m \overline{z_{F_l(m-1)+j}^{(l)}} \tag{103}$$

where, according to equations equation 99 and equation 102,

$$\overline{z_{F_l(m-1)+j}^{(l)}} = \rho \left( \boldsymbol{\Omega}_m \sum_{k=1}^{F_{l-1}} \left( W_{j,k}^{(l)} z_k^{(l-1)} \right) + \boldsymbol{\Phi}_m \right) \tag{104}$$

Hence,

$$\tilde{a}^{(l+1)} = \sum_{j=1}^{F_l} \boldsymbol{W}_{i,j}^{(l+1)} \sum_{m=1}^{\tau} \boldsymbol{C}_m \rho \left( \boldsymbol{\Omega}_m \sum_{k=1}^{F_{l-1}} \left( W_{j,k}^{(l)} z_k^{(l-1)} \right) + \boldsymbol{\Phi}_m \right) \tag{105}$$

which is equal to $a_i^{(l+1)}$ based on equation 96. $\qquad \square$

**Part 2)** Let $\overline{\boldsymbol{B^{(l+1)}}} = \boldsymbol{\Phi} \otimes \boldsymbol{J}_{F_{l+1},1}$. We can define $\overline{a^{(l+1)}}$ as follows:

$$\left[ \overline{a_1^{(l+1)}} \quad \overline{a_2^{(l+1)}} \quad \dots \quad \overline{a_{\tau(F_{l+1})}^{(l+1)}} \right]^{tr} = \boldsymbol{\Omega} \otimes \boldsymbol{a}^{(l+1)} + \overline{\boldsymbol{B^{(l+1)}}}. \tag{106}$$

Therefore, using Equations (94), (95) and (106), we can write

$$\overline{a^{(l+1)}} = \overline{W^{(l+1)}} \, \overline{z^{(l)}} + \overline{\boldsymbol{B^{(l+1)}}} \tag{107}$$

, where

$$\overline{W^{(l+1)}} = \boldsymbol{\Omega} \otimes \left( \boldsymbol{C}^{tr} \otimes W^{(l+1)} \right) = \left( \boldsymbol{\Omega} \otimes \boldsymbol{C}^{tr} \right) \otimes W^{(l+1)}. \tag{108}$$

Moreover, if we define

$$\overline{z_q^{(l+1)}} = \rho \left( \overline{a_q^{(l+1)}} \right) \quad \forall q \in \{1, \dots, \tau(F_{l+1})\}, \tag{109}$$

we can observe that

$$\boldsymbol{z^{(l+1)}} = \left( \boldsymbol{C}^{tr} \otimes \boldsymbol{I}_{F_{l+1}} \right) \overline{\boldsymbol{z^{(l+1)}}}. \tag{110}$$

*Proof.* We know that

$$z_i^{(l+1)} = \rho^*(a_i^{(l+1)}) = \sum_{n=1}^{\tau} \rho\left(\mathbf{\Omega}_n a_i^{(l+1)} + \mathbf{\Phi}_n\right). \tag{111}$$

Now, let us calculate each entry of the RHS of Equation equation 110

$$\left[\left(\mathbf{C}^{tr} \otimes \mathbf{I}_{F_{l+1}}\right) \overline{\mathbf{z}^{(l+1)}}\right]_i = \left[\mathbf{C}^{tr} \otimes \mathbf{I}_{F_{l+1}}\right]_i \overline{\mathbf{z}^{(l+1)}} = \sum_{j=1}^{\tau F_{l+1}} \left(\mathbf{C}^{tr} \otimes \mathbf{I}_{F_{l+1}}\right)_{i,j} \overline{\mathbf{z}_j^{(l+1)}}. \tag{112}$$

Hence, according to equation 90, we have

$$\left[\left(\mathbf{C}^{tr} \otimes \mathbf{I}_{F_{l+1}}\right) \overline{\mathbf{z}^{(l+1)}}\right]_i = \sum_{j=1}^{\tau F_{l+1}} \mathbf{C}^{tr}_{\lceil i/F_{l+1}\rceil, \lceil j/F_{l+1}\rceil} \delta_{1+(i-1)\%F_{l+1}, 1+(j-1)\%F_{l+1}} \overline{\mathbf{z}_j^{(l+1)}}, \tag{113}$$

in which $\delta$ refers to Kronecker delta. As a result,

$$\left[\left(\mathbf{C}^{tr} \otimes \mathbf{I}_{F_{l+1}}\right) \overline{\mathbf{z}^{(l+1)}}\right]_i = \sum_{j=1}^{\tau F_{l+1}} \mathbf{C}_{\lceil j/F_{l+1}\rceil, \lceil i/F_{l+1}\rceil} \delta_{1+(i-1)\%F_{l+1}, 1+(j-1)\%F_{l+1}} \overline{\mathbf{z}_j^{(l+1)}} \tag{114}$$

Note that $1 \le i \le F_{l+1}$. Therefore, $\lceil i/F_{l+1}\rceil = 1$, and $(i-1)\%F_{l+1} = i-1$. Hence,

$$\left[\left(\mathbf{C}^{tr} \otimes \mathbf{I}_{F_{l+1}}\right) \overline{\mathbf{z}^{(l+1)}}\right]_i = \sum_{j=1}^{\tau F_{l+1}} \mathbf{C}_{\lceil j/F_{l+1}\rceil} \delta_{i, 1+(j-1)\%F_{l+1}} \overline{\mathbf{z}_j^{(l+1)}}. \tag{115}$$

Also note that $\delta_{i, 1+(j-1)\%F_{l+1}}$ is zero, except when $j = kF_{l+1}+i$, in which case $\delta_{i, 1+(j-1)\%F_{l+1}} = 1$. Thus,

$$\left[\left(\mathbf{C}^{tr} \otimes \mathbf{I}_{F_{l+1}}\right) \overline{\mathbf{z}^{(l+1)}}\right]_i = \sum_{k=0}^{\tau-1} \mathbf{C}_{\lceil (kF_{l+1}+i)/F_{l+1}\rceil} \overline{\mathbf{z}_{kF_{l+1}+i}^{(l+1)}} = \sum_{k=0}^{\tau-1} \mathbf{C}_{k+\lceil i/F_{l+1}\rceil} \overline{\mathbf{z}_{kF_{l+1}+i}^{(l+1)}}$$

$$= \sum_{k=0}^{\tau-1} \mathbf{C}_{k+1} \overline{\mathbf{z}_{kF_{l+1}+i}^{(l+1)}} = \sum_{n=1}^{\tau} \mathbf{C}_n \overline{\mathbf{z}_{(n-1)F_{l+1}+i}^{(l+1)}} = \sum_{n=1}^{\tau} \mathbf{C}_n \rho\left(\overline{\mathbf{a}_{(n-1)F_{l+1}+i}^{(l+1)}}\right). \tag{116}$$

Note that

$$\overline{\mathbf{a}_{(n-1)F_{l+1}+i}^{(l+1)}} = \mathbf{\Omega}_{\lceil ((n-1)F_{l+1}+i)/F_{l+1}\rceil} \mathbf{a}_{1+((n-1)F_{l+1}+i-1)\%F_{l+1}}^{(l+1)} + \mathbf{\Phi}_{\lceil ((n-1)F_{l+1}+i)/F_{l+1}\rceil}$$

$$= \mathbf{\Omega}_{n-1+\lceil i/F_{l+1}\rceil} \mathbf{a}_{1+(i-1)\%F_{l+1}}^{(l+1)} + \mathbf{\Phi}_{n-1+\lceil i/F_{l+1}\rceil} \tag{117}$$

Since $\left\lceil \frac{i}{F_{l+1}} \right\rceil = 1$ and $(i-1)\%F_{l+1} = i-1$, we have

$$\overline{\mathbf{a}_{(n-1)F_{l+1}+i}^{(l+1)}} = \mathbf{\Omega}_n \mathbf{a}_i^{(l+1)} + \mathbf{\Phi}_n \tag{118}$$

Finally, utilizing Equations equation 116 and equation 118, we deduce that

$$\left[\left(\mathbf{C}^{tr} \otimes \mathbf{I}_{F_{l+1}}\right) \overline{\mathbf{z}^{(l+1)}}\right]_i = \sum_{n=1}^{\tau} \mathbf{C}_n \rho\left(\mathbf{\Omega}_n \mathbf{a}_i^{(l+1)} + \mathbf{\Phi}_n\right), \tag{119}$$

which is equal to the RHS of the Equation equation 110.

**Part 3)** Using parts 1 and 2 of the proof, we can state the theorem for arbitrary even values of $L$. By setting $l = 1$ in the previous parts, we obtain

$$\overline{\mathbf{W}^{(1)}} = \mathbf{\Omega} \otimes \mathbf{W}^{(1)}, \quad \overline{\mathbf{B}^{(1)}} = \mathbf{\Phi} \otimes \mathbf{J}_{F_1, 1} \tag{120}$$

and

$$\overline{\mathbf{W}^{(2)}} = \left(\mathbf{\Omega} \otimes \mathbf{C}^{tr}\right) \otimes \mathbf{W}^{(2)}, \quad \overline{\mathbf{B}^{(2)}} = \mathbf{\Phi} \otimes \mathbf{J}_{F_2, 1}. \tag{121}$$

Thus,

$$\overline{W^{(l)}} = \begin{cases} \Omega \otimes W^{(l)}, & \text{if } l = 1 \\ (\Omega \otimes C^{tr}) \otimes W^{(l)}, & \text{if } l = 2 \end{cases}, \quad \overline{B^{(l)}} = \Phi \otimes J_{F_l,1}. \tag{122}$$

In addition, by setting $L = 2$, we will have $\overline{f_{\overline{\theta}}}(r) = \overline{W^{(3)}}\,\overline{z^{(2)}}$. Note that according to the assumptions of the theorem, $\overline{W^{(3)}} = C^{tr} \otimes I_{F_2}$. As a result, $\overline{f_{\overline{\theta}}}(r) = \overline{W^{(3)}}\,\overline{z^{(2)}} = (C^{tr} \otimes I_{F_2})\,\overline{z^{(2)}}$, which is equal to $z^{(2)} = f_\theta(r)$, as derived in equation 110. equation 110. In conclusion, the theorem holds true for $L = 2$.

Now, suppose that Equation equation 5 holds for $L = 2k$. Consequently,

$$z^{(2k)} = \left(C^{tr} \otimes I_{F_{2k}}\right)\overline{z^{(2k)}} \tag{123}$$

Now, we aim to analyze the case for $L = 2(k+1)$. For this network with two additional layers, we first need to adjust the weight matrix for layer $l = 2k + 1$. The new weight matrix will be

$$\overline{W^{(2k+1)}} = \left(\Omega \otimes W^{(2k+1)}\right)\left(C^{tr} \otimes I_{F_{2k}}\right), \tag{124}$$

and the weights and the biases of the two new layers will be

$$\overline{W^{(2k+2)}} = \left(\Omega \otimes C^{tr}\right) \otimes W^{(2k+2)}, \quad \overline{B^{(2k+2)}} = \Phi \otimes J_{F_{2k+2},1},$$
$$\overline{W^{(2k+3)}} = C^{tr} \otimes I_{F_{2k+2}}, \quad \overline{B^{(2k+3)}} = \Phi \otimes J_{F_{2k+3},1}. \tag{125}$$

Now, note that

$$\overline{W^{(2k+1)}}\,\overline{z^{(2k)}} = \left(\Omega \otimes W^{(2k+1)}\right)\left(C^{tr} \otimes I_{F_{2k}}\right)\overline{z^{(2k)}}. \tag{126}$$

Therefore, by setting $l = 2k - 1$ in Equation equation 110, or using Equation equation 123, we obtain

$$\overline{W^{(2k+1)}}\,\overline{z^{(2k)}} = \left(\Omega \otimes W^{(2k+1)}\right)z^{(2k)} \tag{127}$$

This is analogous to feeding $z^{(2k)}$ into a neural network whose first layer has the weight matrix $\Omega \otimes W^{(2k+1)}$. Since the additional weight matrices and biases are consistent with Parts 1 and 2 of the proof, we can conclude that

$$\overline{f_{\overline{\theta}}}(r) = z^{(2k+2)} = f_\theta(r). \tag{128}$$

$\square$

### D.4 Proof of Lemma equation 4.4

*Proof.* Let $[a_{r,1}, a_{r,2}, \ldots, a_{r,T}] \in \mathbb{Q}^T$ be the $r$'th row of $\Psi^{tr}$. Now, define a matrix $\hat{A}$ which is identical to $A$ except for its $r$'th row. This modified row is constructed as follows:

$$\hat{a}_{r,i} = \frac{\sqrt{p_i}}{10^{-\eta}\lfloor 10^\eta \sqrt{p_i}\rfloor}\left(\psi_{r,i} + \epsilon[\psi_{r,i} = 0]\right) \tag{129}$$

in which $p_i$ is the $i$'th prime number, $\epsilon$ is the machine precision, $[.]$ is Iverson bracket, and $\eta$ is a large enough natural number such that $\frac{\sqrt{p_i}}{10^{-\eta}\lfloor 10^\eta\sqrt{p_i}\rfloor} \approx 1$ (to avoid significant changes in the matrix). At the same time, we must have $|\frac{\sqrt{p_i}}{10^{-\eta}\lfloor 10^\eta\sqrt{p_i}\rfloor} - 1| \geq \epsilon$ (to prevent it from becoming a rational number).

Let $\alpha_i := \frac{\hat{a}_{r,i}}{\sqrt{p_i}}$. Then, $\alpha_i \in \mathbb{Q} \setminus \{0\}$. Now assume that there is $S = [s_1, ..., s_T]^{tr} \in Ker(\hat{A}) \cap \mathbb{Q}^T$. Consequently,

$$\sum_{i=1}^{T} \hat{a}_{r,i} s_i = 0 \tag{130}$$

As a result,

$$\sum_{i=1}^{T} \alpha_i \sqrt{p_i} s_i = 0 \tag{131}$$

Note that $\alpha_i s_i \in \mathbb{Q}$. Furthermore, The square roots of all prime numbers are linearly independent over $\mathbb{Q}$ (Stewart, 2022). As a result, $\alpha_i s_i = 0$ for all $i$. Since $\alpha_i \neq 0$, we must have $s_i = 0$ for all $i$, that is, $Ker(\hat{A}) \cap \boldsymbol{Q}^T = \boldsymbol{O}$.[5] $\qquad\square$

# E   EXACT EXPRESSIVE POWER OF A 2-LAYER STAF NETWORK

In Theorem 4.1, we discussed the set of potential frequencies. Following Novello et al. (2025), one can determine the exact set of frequencies using the Jacobi–Anger expansion. By Theorem 4.2, STAF admits a Kronecker-equivalent sinusoidal network. This allows us to directly apply the following results of Novello et al. (2025).

**Theorem E.1** ((Novello et al., 2025, Thm. 1)). *Each hidden neuron $h_i$ of a 3-layer sinusoidal MLP has an amplitude-phase expansion of the form*

$$h_i(x) = \sum_{\mathbf{k} \in \mathbb{Z}^m} \alpha_{\mathbf{k}} \sin(\beta_{\mathbf{k}} x + \lambda_{\mathbf{k}}),$$

*where $\beta_{\mathbf{k}} = \langle \mathbf{k}, \omega \rangle$, $\lambda_{\mathbf{k}} = \langle \mathbf{k}, \varphi \rangle + b_i$, and*

$$\alpha_{\mathbf{k}} = \prod_j J_{k_j}(W_{ij}),$$

*with $J_k$ the Bessel functions of the first kind.*

**Theorem E.2** ((Novello et al., 2025, Thm. 2)). *The magnitudes of the amplitudes in the above expansion satisfy*

$$|\alpha_{\mathbf{k}}| \leq \prod_{j=1}^{m} \left( \frac{|W_{ij}|}{2} \right)^{|k_j|} \frac{1}{|k_j|!}.$$

In our setting, the Kronecker equivalence established in Theorem 4.2 implies that a 2-layer STAF network corresponds to a 3-layer sinusoidal MLP with weights

$$\omega = \Omega \otimes W^{(1)} \in \mathbb{R}^{\tau F_1 \times F_0}, \qquad W = (\Omega \otimes C_S^{tr}) \otimes W^{(2)};$$

where $\omega \in \mathbb{R}^{m \times d}$. Thus, we have $m = \tau F_1$ and $d = F_0$. By adapting the above theorems to our notation, we obtain

$$\alpha_{\mathbf{k}} = \prod_j J_{k_j} \left( \Omega_{\lceil i/F_2 \rceil, 1} \cdot (C_S)_{\lceil j/F_1 \rceil \% \tau + 1, 1} \cdot W^{(2)}_{i\%F_2, j\%F_1} \right), \tag{132}$$

and the amplitude bound

$$\alpha_{\mathbf{k}} \leq \prod_{j=1}^{\tau F_1} \left( \frac{(C_S)_{\lceil j/F_1 \rceil \% \tau + 1, 1} W^{(2)}_{i\%F_2, j\%F_1}}{2} \right)^{|k_j|} \frac{1}{|k_j|!}. \tag{133}$$

*Proof.* Let $C_S$ denote the matrix $C$ in Theorem 4.2, and $C_T$ denote the matrix $C$ in TUNER's notation. Similarly, we use $W$ (without superscripts or bars) to refer to weights in TUNER, while $\overline{W^{(\cdot)}}$ follows STAF's notation. Then

$$W = \overline{W^{(2)}} = (\Omega \otimes C_S^{tr}) \otimes W^{(2)}, \qquad C_T = \overline{W^{(3)}} = C_S^{tr} \otimes I_{F_2}. \tag{134}$$

As a result, based on E.1, the resulting function can be written as

$$h_i(x) = \sum_{\mathbf{k} \in \mathbb{Z}^m} \alpha_{\mathbf{k}} \sin(\beta_{\mathbf{k}} x + \lambda_{\mathbf{k}}), \tag{135}$$

---

[5]Note that all algebraic numbers are computable. This analysis was founded on the computability and expressibility of the square roots of prime numbers in a machine. However, most of the computable numbers are rounded or truncated when stored in a machine. Nevertheless, it is possible to demonstrate theoretically or through simulation that increasing precision can make the aforementioned analysis always feasible.

where

$$\alpha_{\mathbf{k}} = \prod_j J_{k_j}(W_{i,j}) \tag{136}$$

Note that based on the equation 90,

$$\begin{aligned}
W_{ij} &= \left( (\Omega \otimes C_S^{tr}) \otimes W^{(2)} \right)_{ij} \\
&= (\Omega \otimes C_S^{tr})_{\lceil i/F_2 \rceil, \lceil j/F_1 \rceil} \cdot W^{(2)}_{(i\%F_2)+1, (j\%F_1)+1} \\
&= (\Omega \otimes C_S^{tr})_{\lceil i/F_2 \rceil, \lceil j/F_1 \rceil / \tau} \cdot (C_S^{tr})_{1, (\lceil j/F_1 \rceil \% \tau)+1} \cdot W^{(2)}_{(i\%F_2)+1, (j\%F_1)+1} \\
&= \Omega_{\lceil \frac{i}{F_2} \rceil, 1} \cdot (C_S)_{\lceil \frac{j}{F_1} \rceil \% \tau+1, 1} \cdot W^{(2)}_{i\%F_2, j\%F_1} \tag{137}
\end{aligned}$$

Consequently, according to equation 136 and equation 137,

$$\alpha_{\mathbf{k}} = \prod_j J_{k_j}\left( \Omega_{\lceil \frac{i}{F_2} \rceil, 1} \cdot (C_S)_{\lceil \frac{j}{F_1} \rceil \% \tau+1, 1} \cdot W^{(2)}_{i\%F_2, j\%F_1} \right). \tag{138}$$

In addition, the frequency term can be written as

$$\beta_{\mathbf{k}} = \langle \mathbf{k}, \omega \rangle = \langle \mathbf{k}, \Omega \otimes W^{(1)} \rangle = \sum_{\ell=1}^{m} k_\ell \, (\Omega \otimes W^{(1)})_\ell. \tag{139}$$

Based on Theorem E.2, the amplitudes satisfy

$$\alpha_{\mathbf{k}} \le \prod_{j=1}^{m} \left( \frac{|W_{ij}|}{2} \right)^{|k_j|} \frac{1}{|k_j|!}. \tag{140}$$

Therefore, using equation 137,

$$\alpha_{\mathbf{k}} \le \prod_{j=1}^{\tau F_1} \left( \frac{|\Omega_{\lceil i/F_2 \rceil, 1} (C_S)_{\lceil j/F_1 \rceil \% \tau+1, 1} W^{(2)}_{i\%F_2, j\%F_1}|}{2} \right)^{|k_j|} \frac{1}{|k_j|!}. \tag{141}$$

Grouping terms, we have

$$\prod_{j=1}^{\tau F_1} \left( (C_S)_{\lceil j/F_1 \rceil \% \tau+1, 1} \right)^{|k_j|} = \prod_{i=1}^{\tau} (C_S)_{i,1}^{\sum_{j=1}^{\tau F_1} |k_j|}. \tag{142}$$

Therefore,

$$\alpha_{\mathbf{k}} \le \left| \Omega_{\lceil i/F_2 \rceil, 1} \prod_{i=1}^{\tau} (C_S)_{i,1} \right|^{\sum_{j=1}^{\tau F_1} |k_j|} \prod_{j=1}^{\tau F_1} \left( \frac{|W^{(2)}_{i\%F_2, j\%F_1}|}{2} \right)^{|k_j|} \frac{1}{|k_j|!} \tag{143}$$

$$\square$$

# F  AN ALTERNATIVE PROOF OF THE ASYMPTOTIC BEHAVIOR OF DELANNOY NUMBERS

*Proof.* Let $V(A, B)$ denote a Delannoy number. This number represents the number of lattice points inside and on the $L1$ sphere (or equivalently, the cells in the von Neumann neighborhood) in $A$ dimensions with radius $B$. We also know that one of the properties of these numbers is symmetry with respect to the arguments Kiselman (2012); that is, for any $A$ and $B$, we have:

$$V(A, B) = V(B, A). \tag{144}$$

Hence,

$$\frac{|V(\tau T, K)|}{|V(T, K)|} = \frac{|V(K, \tau T)|}{|V(K, T)|}. \tag{145}$$

The second ratio means that if the radius of the sphere is scaled by $\tau$, the number of lattice points inside and on the $L1$ sphere is multiplied accordingly. It is known that the number of lattice points can be approximated using the volume of the sphere. Moreover, when the radius is scaled by $\tau$ in $K$ dimensions, the volume of an object is multiplied by $\tau^K$. Consequently, the number of lattice points also grows approximately by a factor of $\tau^K$. Therefore,

$$\frac{|V(\tau T, K)|}{|V(T, K)|} \sim \tau^K. \tag{146}$$

$\square$

## G  DISCUSSION AND LIMITATIONS

**When does STAF help?** Tasks whose target signals contain mixed frequencies or repetitive fine detail benefit most, especially when positional encodings are absent.

**Tuning.** Very large $\tau$ increases compute and may slow late-phase convergence. We recommend small $\tau$ with layer-wise sharing and the unit-variance initialization.

**What we do not claim.** We do not claim to eliminate spectral bias or to dominate all baselines on all metrics. Our results indicate consistent improvements in convergence and fidelity across diverse settings, with clear but bounded limitations.

