# OpenReview forum: "A Unified Theory of Sinusoidal Activation Families for Implicit Neural Representations"
_ICLR.cc/2026/Conference — ICLR 2026 Conference Withdrawn Submission_

### Official Review · Reviewer_5heZ · 2025-10-29

**Soundness:** 2
**Presentation:** 2
**Contribution:** 2
**Rating:** 2
**Confidence:** 5

**Summary:**

This paper seeks to propose a general theoretical and practical framework for studying sinusoidal activations within the context of implicit neural representations (INRs). Their approach is to see a sinusoidal activation, as for example used in the SIREN work by Sitzmann et al., as only one piece of a broader story that employs a Fourier series activation. In the authors framework a Fourier series activation is expressed in the usual way, as a sum of sinusoids, but where the frequency, amplitude and phase is learnt during the optimization of the network. Their theoretical analysis exhibits a construction that allows one to view these networks using Kronecker products, establishes a characterization of the spectrum of the NTK and provides an initialization for such networks that does not require the central limit theorem in their argument. They then empirically validate the performance of their networks on a variety of INR tasks.

**Strengths:**

**Clarity:** The authors clarify the roll of sinusoidal activations within the context of INRs well and explain some of the issues and problems that such an activation bring. In this regard they have done a good job. For example, on lines 223-230 they pay particular attention to the initialization carried out in SIREN by Sitzmann et al., and point out that there are several assumptions used in the theoretical argument given in the SIREN paper. The authors' initialization takes a different route which circumvents many of the approximations used by Sitzmann et al, which makes their initialization much more intuitive. Furthermore, the authors make the first steps in calculating the explicit NTK of their networks which is something that most INR papers don't do. However, I was a bit underwhelmed by their computation as I could not understand what their purpose in calculating the NTK was. In appendix A.1 the calculation is presented yet with no information on what that is telling the reader. Is this supposed to be compared with the NTK of other INRs to show something theoretical?

**Weaknesses:**

**Novelty:** While the paper does seek to better understand the role of periodic activations within the context of INRs I don't see any real novelty that would push the INR community forward. It seems to me that the authors have just analysed what happens when they take a sum of sinusoids as activations and have developed a framework that encompasses this type of activation while in the process showing that methods such as SIREN fit into this framework. What is the actual novelty of this approach?

**Significance:** This is related to my novelty comment made above. As the paper develops a framework for Fourier series as activations I am feeling like this is just another paper that is proposing yet another activation for the INR community and it seems that this activation is really just a sum of sinusoids. I do admit their method does yield better performance as shown in their experiments section but I noticed the performance, for example see Table 1, is only slightly better than some of the other methods used in the literature. I am also not understanding in what way the theoretical analysis given in the paper is significant. If the authors could elaborate on this I think that would help.

**Quality:** At times I felt the quality of the paper, especially its writing, is really lacking. The authors seem to say things that are rather on the confusing side of things and just don't make sense. For example, in the introduction from line 40-44, the authors state that "..follow-up work explored alternative periodic bases (e.g. Gabor-like (Saragadam et al., 2023))". The Saragadam et al. paper is WIRE where they use a Gabor filter as an activation. However, the Gabor filter is not periodic. It has exponential decay and form looking at the WIRE paper they, in the real case, use a exponential multiplied by a sinusoid. How is that periodic? Furthermore, in figure 1 the authors show a picture of the Gabor filter and it is not periodic. Another issue I had was that the authors in their abstract end by speaking of the capacity-convergence trade-off. What is the capacity-convergence trade off? Is this that in order to get good convergence you need to increase the capacity of a network? You do realise this is not a trade-off per say but rather an implication of the NTK theory that you in fact reference. In the infinite width limit, capacity essentially approaches infinity, and that's where the NTK regime kicks in and get good convergence. So I am confused with what you are trying to say. In that same sentence you say "trainable sinusoids reshape the optimization landscape...". Yet once again I am left confused. In what way do trainable sinusoids reshape the optimization landscape? On lines 223-230 you point out some issues with the approach taken by SIREN for their initialization which I think was very good. However, it would have been much better had you clearly stated what those approximations the SIREN paper used, why they are not good, and then clearly related them to your initialization showing that you don't need those approximations. This would have put your initialization in much stronger standing. I suggest you include this in the appendix. I was also completely confused with Theorem 4.3. It show that STAF can increase the size of the potential frequencies is that correct? But why is this useful for learning? It could be that the extra frequencies are not present in the signal you are trying to reconstruct. Also, I feel this is not a surprising statement as STAF uses a some of sinusoids whose phase, frequency, and amplitude are being learnt so you would expect it to be able to have capacity to learn more frequencies in the target signal. It could be I am missing something here. I also noticed that you didn't really speak about any of the limitations of your approach.

**Questions:**

1. In the abstract what do you mean by the capacity-convergence trade-off? This seems to go against the obvious implication of the NTK theory at the infinite width limit.

2. You also mention at the end of the abstract that trainable sinusoids reshape the optimization landscape. Could you please explain what you mean? Is it the case by using trainable sinusoid the curvature of the loss landscape changes? Is this what you mean?

3. What is the point of Theorem A.2? What exactly is this theorem supposed to be telling me?

4. In lines 40-44 you seem to suggest that Gabor-like bases are periodic. However, they are not from what I know. Furthermore, in figure 1 you show a Gabor base and it is not periodic. Can you please explain what you meant in those lines?

5. I fail to see what is the novelty and significance of this paper. It seems to me that you are using an activation that is simply a sum of sinusoids, see eqn. (2) of the paper. And that because you are using a sum that somehow these activations are more representable. Is this the main novelty?

6. What are the limitations of your STAF framework? By modelling the activations using Fourier series aren't you assuming the target signal must be periodic? This was an issue with SIREN. So I would imagine if you had test points that were far away from the training distribution, and the target signal was not periodic, you would get bad generalization. The reason you don't see this in the experiments is because INRs are generally hypersampled so the network is pretty much memorizing.

---

### Official Review · Reviewer_weoE · 2025-10-31

**Soundness:** 3
**Presentation:** 2
**Contribution:** 3
**Rating:** 4
**Confidence:** 4

**Summary:**

This paper extends the SIREN architecture by generalizing its activation function from a single sine to a sum of sines. In each layer, the activation is defined as a weighted sum of sine functions with learnable parameters. This design allows the model to represent a broader range of frequencies than the original SIREN while introducing only a modest increase in the number of parameters.

**Strengths:**

The method is theoretically grounded, leveraging probabilistic reasoning and Neural Tangent Kernel (NTK) analysis.

The paper includes extensive experimental results in the appendix.

The proposed idea is conceptually simple yet potentially powerful. However, the overall presentation could be improved. The paper would benefit from better organization. In particular, several results in the appendix could be moved to the main text to better highlight the advantages of the proposed approach.

**Weaknesses:**

[L52] The first bullet point in the contributions section is unclear. The novelty lies in using a sum of sines as the activation, of which SIREN is a special case with a single sine.

[Sec. 3.4] The authors describe three variants of the proposed activation. Please include clear references or pointers to the corresponding ablation results.

[L98–103] It is not clear whether this discussion refers to neural networks beyond the INR setting. If so, it could be omitted, as the vanishing-gradient issue for SIRENs does not occur when using the initialization proposed by Sitzmann et al.

[L216–230] These paragraphs are more suitable for an appendix. The main paper could focus more on the method and its evaluation.

The appendix organization also needs improvement. For example, Section B2 references the Tokyo image (p. 19), but the corresponding figure appears only on p. 24. Tables 3–4 are cited on p. 20 but appear after several figures on p. 23, which makes navigation difficult.
[Figs. 8 and 9] For the denoising and super-resolution tasks, the improvement over the second-best baseline is minimal (< 1 dB). Including statistical analyses (e.g., mean ± std over multiple runs) would strengthen the results.

[L1727] Contains a duplicated “equation” word.

**Questions:**

[Fig. 3, right] The training of STAF appears unstable and does not seem to converge. What happens if the model is trained for more iterations?

[Fig. 5] Qualitative improvements appers minor. Consider adding absolute error maps or analytical gradient visualizations (as in TUNER) for clearer comparison.
[Fig. 6] Including numerical mean errors or frequency-domain visualizations (e.g., spectrograms) could make the analysis more insightful.

[Table 2] The PSNR values across scenes are relatively low, even though STAF performs best among the compared methods. In FINER’s paper, the reported values are higher. Could the authors comment on this discrepancy?

Overall, Tables 3 and 4 seem to better illustrate the proposed method’s advantages and might be more appropriate for inclusion in the main text than Table 2.

---

### Official Review · Reviewer_qK4B · 2025-10-31

**Soundness:** 3
**Presentation:** 3
**Contribution:** 3
**Rating:** 6
**Confidence:** 2

**Summary:**

This paper “A Unified Theory of Sinusoidal Activation Families for Implicit Neural Representations” proposes STAF, a trainable sinusoidal activation with learnable amplitudes , frequencies , and phases and combines it in a theoretical framework that shows (i) how trainable sinusoids expand the frequency set compared to SIREN, (ii) how this shows up in the NTK spectrum, and (iii) how to initialize such networks. Experiments are shown on images, audio, shapes, inverse problems, and NeRF.

**Strengths:**

1). The paper is written well


2). The mathematical rigor and theoretical support for the proposed method are very strong.


3). STAF introduces a mathematically grounded initialization scheme for sinusoidal activations that replaces SIREN’s heuristic.


4). The paper presented extensive set of experiments on most inr related tasks with a good set of comparison baselines.

5). I have seen often existing INRs make color changes to the reconstructed image, but it seems the STAF helps to reduce this effect (Figure 12). I suggest authors to, if possible, add some section regarding this color preservation in your final version as STAF seems to
minimize this effect. This could be due to optimized sinusoids rather than fixed activations like single sinusoid.

**Weaknesses:**

1). INRs have a wide range of applications for instance, in video representation, signal compression, medical imaging, change detection, etc. However, none of these domains are discussed in the related work. I suggest that the authors include these diverse INR application domains, incorporating the latest works.

2). The novelty of the activation is a bit limited as making the amplitude, phase, and frequency learnable is straightforward.

3). The performance improvement might primarily arise from the Fourier-like additive structure of multiple sinusoids and the inclusion of learnable parameters for each component.

4). Even though the authors presented many results on signal overfitting examples, when it comes to inverse vision tasks, a limited number of examples are shown (one for super resolution and one for denoising). I suggest authors to show more examples on these along with image inpainting (no need to show on entire datasets, 3-4 examples would be sufficient).

5). In the denoising experiment, τ is set to 2. Is this setting based on a single example, or averaged across a dataset? Also, does τ=2 generalize across different noise types? Similarly, in the image overfitting tasks, does τ vary with changes in the number of hidden
neurons or hidden layers?


6). The paper provides limited intuitive interpretation of how the learned sinusoidal components (amplitudes, frequencies, and phases) evolve during training or how they correlate with signal characteristics. If possible, authors can try a signal with a few known frequency components and show how the STAF’s training dynamics adjust the trainable parameters to fir the known frequency components.

**Questions:**

See the weaknesses section

---

### Official Review · Reviewer_mZcj · 2025-11-01

**Soundness:** 2
**Presentation:** 3
**Contribution:** 2
**Rating:** 4
**Confidence:** 3

**Summary:**

The paper presents an activation framework for implicit neural networks dubbed STAF, where they use a combination of sine activations per layer. They propose to learn the parameters of the MLP and the activations simultaneously and claim that this leads to superior performance. They also provide several theoretical results including a Kronecker-equivalence construction that expresses trainable sinusoidal activations with standard sine networks, insights into the NTK of the network, and also propose an initialization scheme for the proposed activation function. The experiments cover images, audio, and 3D shape reconstruction.

**Strengths:**

The quantitative results of the proposed activation function seems to be compelling across different data modalities. I also liked the section where the authors point out the short comings of the analysis in SIREN. The proposed initialization scheme also seems to be well grounded. Overall the paper is well-written and is easy to read.

**Weaknesses:**

All the major theoretical results of the paper seem to be heavily reliant on existing work. This, in my opinion, limits the contribution of the paper. For instance, the major theoretical result of the paper, Theorem 4.2, is a special case of the analysis provided in Jagtap et al. Also, the NTK analysis is also a direct substitution of variables into existing results in the literature. Since the paper presents these theoretical results as major contributions of the paper, I believe such heavy dependency limits the novelty of the analysis. I invite the authors to correct me if I am wrong about this.

**Questions:**

Not sure what Fig 2 is supposed to conclude. I cannot see anything special about the proposed method there. Can the authors please clarify this?

How does your initialization scheme, in theory, compare against the method proposed in [1]?

The authors say "If the learning process is well-posed and there is sufficient data, the training process is likely to converge to a stable and
accurate solution. Therefore, while it is important to monitor potential issues in later epochs, the concern about vanishing or exploding values is significantly greater during the initial stages.". I'm not sure how would someone come to such a conclusion. Can the authors clarify this?

Only one qualitative example per experiment is given for 3D shape representation and NeRF. This is not enough to validate the efficacy of the method over these tasks.

[1] - VI3NR: Variance Informed Initialization for Implicit Neural Representations

---

### Note · Authors · 2025-11-17

I have read and agree with the venue's withdrawal policy on behalf of myself and my co-authors.